# A Novel 3D Reconstruction Sensor Using a Diving Lamp and a Camera for Underwater Cave Exploration

**DOI:** 10.3390/s24124024

**Published:** 2024-06-20

**Authors:** Quentin Massone, Sébastien Druon, Jean Triboulet

**Affiliations:** Laboratory of Informatics, Robotics and MicroElectronics (LIRMM) (UMR 5506 CNRS—UM), University of Montpellier, 161 Rue Ada, CEDEX 5, 34392 Montpellier, France; quentin.massone@gmail.com (Q.M.); sebastien.druon@lirmm.fr (S.D.)

**Keywords:** structured light method, active 3D reconstruction, cone estimation, underwater karst aquifer

## Abstract

Aquifer karstic structures, due to their complex nature, present significant challenges in accurately mapping their intricate features. Traditional methods often rely on invasive techniques or sophisticated equipment, limiting accessibility and feasibility. In this paper, a new approach is proposed for a non-invasive, low-cost 3D reconstruction using a camera that observes the light projection of a simple diving lamp. The method capitalizes on the principles of structured light, leveraging the projection of light contours onto the karstic surfaces. By capturing the resultant light patterns with a camera, three-dimensional representations of the structures are reconstructed. The simplicity and portability of the equipment required make this method highly versatile, enabling deployment in diverse underwater environments. This approach is validated through extensive field experiments conducted in various aquifer karstic settings. The results demonstrate the efficacy of this method in accurately delineating intricate karstic features with remarkable detail and resolution. Furthermore, the non-destructive nature of this technique minimizes disturbance to delicate aquatic ecosystems while providing valuable insights into the subterranean landscape. This innovative methodology not only offers a cost-effective and non-invasive means of mapping aquifer karstic structures but also opens avenues for comprehensive environmental monitoring and resource management. Its potential applications span hydrogeological studies, environmental conservation efforts, and sustainable water resource management practices in karstic terrains worldwide.

## 1. Introduction

Access to water has become one of the most pressing challenges facing humanity today. However, water is abundant on a global scale; the oceans are an inexhaustible reserve, provided that salt and water are separated. Unfortunately, desalination methods are still too energy-intensive to be used profitably on a large scale. Pending a possible scientific breakthrough, people are therefore relying essentially on freshwater resources, which account for 2.5% of the world’s water [1] (Figure 1). Only 1.2% of this freshwater is accessible at the surface, although it is now very polluted. Of the remainder, 68.7% is trapped in glaciers and ice caps and 30.1% is found underground. This groundwater, which represents 0.76% of the world’s resources, is extremely valuable. Groundwater is the result of a long and natural process of filtration through the various layers of soil, which means that its quality is generally considered to be very good [2,3]. This groundwater is stored in aquifers, most of which are karstic.

Aquifer karstic structures, due to their complex and porous nature, present significant challenges in accurately mapping their intricate features. Traditional methods often rely on invasive techniques or sophisticated equipment, limiting accessibility and feasibility. In this paper, a novel non-invasive approach is proposed utilizing structured light projection in conjunction with a camera and a diving light to effectively map aquifer karstic formations. Section 2 presents some related works in underwater mapping. Section 3 details the materials and methods used in this original approach based on the principles of structured light, leveraging the precise projection of light contours onto the karstic surfaces. It also outlines our process of capturing the resultant light patterns with a camera (reflex Nikon D700 with underwater housing), leading to reconstruct detailed three-dimensional representations of the subsurface structures. Section 4 presents the results to validate this approach via the simulation of an aquifer galley first, and finally through extensive outdoor experiments conducted in a particular equivalent aquifer karstic environment. The results demonstrate the efficacy of this method in accurately delineating intricate karstic features with remarkable detail and resolution. Furthermore, the non-destructive nature of this technique minimizes disturbance to delicate aquatic ecosystems while providing valuable insights into the subterranean landscape. This innovative methodology not only offers a cost-effective and non-invasive means of mapping aquifer karstic structures but also opens avenues for comprehensive environmental monitoring and resource management. Its potential applications span hydrogeological studies, environmental conservation efforts, and sustainable water resource management in karstic terrains worldwide.

## 2. Related Work

Underwater mapping is a valuable tool in a wide range of applications, including marine biology, geology, archaeology, and the offshore industry. Various methods and sensors are employed to obtain 3D reconstructions of the environment, with data typically collected by divers, ROVs ( Remotely Operated Vehicle), or AUVs ( Autonomous Underwater Vehicle). In most cases, this involves performing surveys with downward-facing sensors and reconstructing the seafloor surface [4,5]. The studies conducted by the authors of [6,7] provide comprehensive reviews of the various underwater 3D reconstruction methods. This synthesis of methods is illustrated in Figure 2.

Sensors used for 3D reconstruction can be classified into two main categories:Active sensors: these sensors physically interact with the environment by emitting a signal or wave, such as sound waves in the case of sonars or light waves in the case of LiDAR systems.Passive sensors: these sensors do not modify the environment; the most commonly used ones for 3D reconstruction are camera-based.

Time-of-flight methods are the most direct methods for 3D reconstruction. They work on the same principle as a rangefinder, namely emitting a signal or wave and retrieving its echo. By knowing the speed of the signal thanks to the nature of the wave and the propagation medium, it is possible to proportionally deduce the relative distance between the sensor and the observed element from the measurement of the time elapsed between the emission of the wave and the return of the echo.

Among time-of-flight methods, those based on acoustic waves are by far the most used because their propagation is favored by the medium. They offer an interesting coverage capacity in front of the size of the areas to be studied in underwater mapping.

In addition, and to allow the localization of 3D data during the measurement, most underwater navigation algorithms are based on acoustic positioning systems such as Doppler Velocity Log (DVL) and Ultra-short baseline (USBL) [8,9]. On the other hand, Multibeam sonars (MBS) are adapted to bathymetric mapping (measurement of the seabed relief) [10]; however, they are much less suitable for structured (non-flat) environments such as karst aquifers.

Gary et al. [11] and Kantor et al. [12] presented a 3D model of the Zacatón cenote (a water-filled cave at least 300 m deep) using sonar data collected by the DEPTHX (Deep Phreatic THermal eXplorer) vehicle. It is a NASA autonomous underwater vehicle measuring just over 2 m and weighing over a ton, prefiguring among the robots that could explore underwater environments on other planets. Its sensors consist of a DVL, a sonar, and an IMU (Inertial Measurement Unit); additionally, its movements are ensured by six motors allowing it to move in all directions.

Another method using acoustics is described by the authors of [13], which shows the results of an AUV performing limited penetration inside an underwater cave using a mechanically scanned imaging sonar (MSIS, namely mechanically scanned imaging sonar). The results, obtained only for localization, show that it is possible to use this type of technology in karst environments.

Though they are efficient for obtaining bathymetry and macro-characteristics of habitats, these sensors nevertheless provide little information on fine characteristics (micro-bathymetry), do not allow for the recovery of the texture of the environment, and are very expensive.

Still in the class of time-of-flight methods, those based on electromagnetic waves, such as LiDAR, offer access to a very dense reconstruction of the seabed. However, on the other hand, they are reserved for very local observations, because the propagation of these waves are limited by the medium as explained in the constraints section. They are therefore not adapted to the situation envisaged.

The other class of methods are triangulation methods which are favored in this presentation. These methods require at least two devices (or two different views) which will provide distance information from the same point, and thus form a triangle between the two devices and the point to be measured.

There are several ways to perform triangulation:Multi-view triangulation involves taking multiple views of the same scene. By knowing the disparities of each viewpoint, it is possible to project the points of the scene onto different “image” planes, and thus triangulate the points of the scene to obtain a depth map. This technique is based on epipolar geometry (explained in [14]) with two possible approaches:–The use of a binocular stereo system where the relative positions of the two cameras are known. It is possible to make the system active by using a structured light projector to add discriminant points or features related to the observed environment, and thus facilitate the matching of points between the two images for triangulation. In [15], an original stereovision system is presented carried by divers for pipeline measurements; however, its performance needs to be evaluated.–The method known as Structure from Motion (SFM) involves taking a sequence of sequential images of an object or scene, typically from a single camera. To obtain a metric reconstruction, it is necessary to know the camera trajectory to resolve the scale ambiguity.Structured light triangulation relies on an emitter-receiver system. The emitter is a light source (laser or not) that projects a pattern (distinctive patterns) onto a scene observed by the receiver—a camera. Using the positions of the camera and the projector, it is possible to triangulate the discriminant points brought by the light in the scene captured by the camera [16]. In [17], results obtained on small objects (spheres) with a measurement distance of about 1 m lead to radii estimation of less than 0.5 mm. In [18], surface measurements based on the fringe projection technique are presented. They describe an extended camera model which takes refraction effects into account. The results have a resolution of less than 150 μm; the study was performed on small objects and needs to be conducted on bigger scenes.

Regarding structured light triangulation methods, they are very accurate, but are generally only used to reconstruct small areas or objects. This is because light is quickly absorbed in water; therefore, the sensor must be close to the target to best detect the projected light. Additionally, structured light sensors are not yet well-suited for mobility and are therefore mostly used statically for the moment.

Digital reflex cameras, on the other hand, have the advantage of being inexpensive, providing density as well as textural data. However, most camera-based methods operate in open areas with natural lighting (or with artificial lighting that completely illuminates the workspace) and generally try to reconstruct the ground or small objects.

For example, refs. [19,20] propose a 3D reconstruction method based on a stereo pair for archaeological objects in shallow water and in very poor conditions. Special filters are implemented to solve the noise problems caused by these poor conditions. This static 3D measurement is fused with a 3D map of the excavation area obtained using a scanning sonar. Brandou et al. [21] performed a dense reconstruction of submerged structures at very high depths (up to 6000 m) with a stereo system mounted on a controlled manipulator arm. The imposition of a known trajectory allows one to know the different positions of the cameras to calculate a precise 3D model of the scene.

Beall et al. [22] reconstructed coral reefs with a wide-baseline stereo system using high-resolution video and a dense reconstruction method.

The context of this paper is different from these works since our experiments are in total darkness in a structured and extended environment whose content appears as the diver or robot moves.

Zhou et al. [23] presented a description of the problems related to metrology in the underwater environment. The importance of calibrating the stereo pair underwater is emphasized, taking into account the refraction problems related to the medium. The detection of interest points is performed on sufficiently bright images allowing a robust matching then used for a 3D reconstruction. The comparison of the object measurements obtained is made with a reference metrology and gives good results; however, only in static cases.

The work of [24] is directly related to the present theme. The objective is to reconstruct a karstic underwater tunnel of *Woodville Karst Plain* in Florida using a new approach to stereovision generating a 3D point cloud of the cave. In their method, after calibration and rectification of the highly distorted images, they extract the contours of the artificial light projected into the cave which creates a cone of light illuminating part of the walls. These contours are then used as inputs in their stereovision algorithm. The displacement is estimated by visual odometry (ORB-SLAM ( Oriented FAST and Rotated BRIEF-Simultaneous Localization and Mapping) [25]) and allows a 3D reconstruction of a portion of the cave of about one hundred meters; however, it does so without specifying the comparison of the result with a known ground truth.

For an even more optimal mapping of this underwater gallery, ref. [26] developed a SLAM method where they fuse this stereo method with a sonar and an inertial measurement unit, thus improving the accuracy of their measurements. Reviews on SLAM methods in underwater environment can be found in refs. [27,28].

As stated by Joshi et al. [29] and Ende [30], the underwater environment is very specific and induces new challenges both in exploring and mapping tasks. The interested reader can refer to the work of Han, Hao and Qi [31] or Papp et al. [32] for an example on how light refraction can impact photogrametric approaches, or to Hou et al. [33] to see how the camera calibration is altered.

Of course, 3D reconstructions have not been preserved from the current trend of deep learning techniques. Ju et al. outlined a good state of such methods for calibrated approaches in ref. [34], but the work of Abdullah et al. [35] is more specific to the semantics of underwater scenes.

Wu et al. [36] used a SFM method coupled with multi-view stereo photography to perform measurements of silting areas at the outlet of water transport pipelines to the seabed. The accuracy of their method allows a robust estimation of these elements useful for monitoring these areas, but it also only uses static measurements.

Fan et al. [37] provided an overview of calibration and underwater image processing methods in structured light. Wang et al. [38] proposed a method of 3D reconstruction in underwater environments combining a SLAM approach and dense reconstruction using a stereo system embedded on the Aqua2 robot whose IMU allows for localization. The results are mainly on coral reefs and partially on the reconstruction of underwater cavities. The question of the accuracy of the measurements with respect to a ground truth remains. Braeuer-Burchardt et al. [39] proposed an application of 3D reconstruction for underwater archeology. Their system consists of two monochrome cameras and a sinusoidal light pattern projector. A combination of 3D reconstruction in structured light and SLAM-based visual odometry is proposed. Their design is mounted on an ROV whose IMU data complements that of visual odometry. The first results were obtained on objects in a controlled environment. A test was then carried out over a dozen meters to follow a pseudo pipeline, but the size of the system was too large for usable application in underground galleries.

Yu et al. [40] used Ariane’s thread to enable the movement and guidance of a BlueRov robot. This learning method gives good results for localization and assisted piloting. The next steps envisaged are autonomous navigation and then mapping of the cavities studied (but were already explored by divers).

Hu et al. [41] proposed a real-time monocular 3D reconstruction method for underwater environments. It relies on optical flow tracking of characteristic seabed points and uses Delaunay triangulation to complete the seabed estimation. The method assumes a nearly planar seabed but remains unclear about the achieved metric accuracy.

## 3. Materials and Methods

The complete development of the proposed method is detailed in Figure 3 and is based on three main blocks:Zhang’s **camera calibration** method [42] is used to estimate its intrinsic parameters, including radial distortion;**light projector calibration** consists of estimating the geometric parameters of the cone (vertex $O_{P}$, direction d, half-angle of opening α);**three-dimensional local reconstruction** is based on the camera/projector triangulation leading to a 3D point cloud expressed in C, the camera reference frame.

This triangulation is illustrated in Figure 4. It shows the system, with one camera and light projector. To simplify the figure, the light is projected onto a plane, but one can imagine that the procedure has to be applied to non-planar surfaces such as the walls of a gallery. Once the light contour has been detected in the image, the associated camera ray is calculated for each point on the contour. Figure 4 shows two points, $X^{\prime }$ and $Y^{\prime }$, belonging to the contour detected in the image and the associated rays. In the configuration shown in Figure 4, the ray associated with $X^{\prime }$ and $Y^{\prime }$, respectively, intersects the cone at two points $X_{1}$ and $X_{2}$, and $Y_{1}$ and $Y_{2}$, respectively. To calculate these intersections, one needs to know the angle of aperture α and the transformation between the references of the cone and the camera called $T_{\left\{P\to C\right\}}$.

Also in Figure 4, the intersections which correspond to the 3D points of the contour ($X_{1}$ and $Y_{2}$) are shown in green and those which should not be taken into account ($Y_{1}$) are shown in red. The point $X_{2}$ is not shown, but one can imagine that it would be red and beyond the area of the plane in Figure 4. For a given ray, it is therefore necessary to identify whether it is the first intersection or the second intersection that belongs to the observed light contour.

This method of selecting intersections requires a study of the geometric relationships between the cone and the camera.

### 3.1. Cone Parameters

#### 3.1.1. Cone of Revolution

The cone of revolution, referred to as a cone in the remainder of this document, is associated with its surface of revolution and not with its solid, as may be the case in certain applications. The terms used to define the relationship between a point and the cone are as follows:a point belonging to the cone is a point which belongs to the surface of the cone;a point inside the cone is a point which belongs to the solid bounded by the surface of the cone;a point outside the cone is a point which does not belong to the solid bounded by the surface of the cone.

This surface is generated by the revolution of a line, called the generatrix, noted g and which passes through a vertex, in this case $O_{P}$.

The revolution takes place around a fixed axis which also passes through the vertex and which turns out to be an axis of symmetry of the cone, the direction of which is defined by the unit vector d.

If a plane not passing through $O_{P}$ is orthogonal to the axis of symmetry, then its intersection with the cone is a circle.

Thus, the angle formed by g and the axis of symmetry is constant and corresponds to the half-angle of the cone opening $\alpha \in ]0;\frac{\pi }{2}[$.

The cone is represented in Figure 5 with the orthonormal base $\left(O_{P},\;x_{P},\;y_{P},\;z_{P}\right)$, oriented such that $z_{P}=d$, which defines the P reference frame associated with the cone.

The generatrix g has the unit direction vector w which is set by a new angle θ∈[0;2π[ (Figure 5).

In the cone frame P, one can simply express the vector $w_{\left\{P\right\}}$ (Equation (Equation 1)) which then depends only on the angles θ and α:(1)$$w_{\left\{P\right\}}=\left[\begin{array}{c} \sin(\alpha )\cos(\theta )\\ \sin(\alpha )\sin(\theta )\\ \cos(\alpha ) \end{array}\right]$$

This vector can be used to express all the points on the surface parametrically.

Let $X_{\left\{P\right\}}={\left[\begin{array}{ccc} x_{X_{\left\{P\right\}}} & y_{X_{\left\{P\right\}}} & z_{X_{\left\{P\right\}}} \end{array}\right]}^{T}$ be a point X expressed in the reference frame P. Then, if X belongs to the cone, one can calculate (Equation (Equation 2)):(2)$$X_{\left\{P\right\}}=lw_{\left\{P\right\}}\Leftrightarrow \left\{\begin{array}{c} \begin{array}{c} x_{X_{\left\{P\right\}}}=l\sin\left(\alpha \right)\cos\left(\theta \right)\\ y_{X_{\left\{P\right\}}}=l\sin\left(\alpha \right)\sin\left(\theta \right)\\ z_{X_{\left\{P\right\}}}=l\cos\left(\alpha \right) \end{array} \end{array}\right.$$
where *l* is a real parameter whose absolute value defines the distance between $X_{\left\{P\right\}}$ and the cone vertex. However, with l∈R, the parametric equation defines a “double cone”. However, since the cone models the light projector, the study is only interested in the upper part of the “double cone”, i.e., when the parameter $l\in R^{+}$.

#### 3.1.2. Orthogonal Distance

This parameterization of the cone will be useful to write the equation for the orthogonal distance between a point and the cone, an equation that will be useful for cone estimation. Let M be a point outside the cone and H be its orthogonal projection on the cone (Figure 5). The angle $\theta_{H}$ is the angle that gives the generatrix of the cone closest to M. This generatrix, called $g_{H}$, passes through H and has the unit direction vector $w_{H}$ whose expression in the P reference frame is:$${w_{H}}_{\left\{P\right\}}=\left[\begin{array}{c} \sin\left(\alpha \right)\cos\left(\theta_{H}\right)\\ \sin\left(\alpha \right)\sin\left(\theta_{H}\right)\\ \cos(\alpha ) \end{array}\right]$$

If M is also expressed in P with $M_{\left\{P\right\}}={\left[\begin{array}{ccc} x_{M_{\left\{P\right\}}} & y_{M_{\left\{P\right\}}} & z_{M_{\left\{P\right\}}} \end{array}\right]}^{T}$, the expressions become (Equation (Equation 3)):(3)$$\left\{\begin{array}{c} \begin{array}{cc} \theta_{H} & =\arctan\left(\frac{y_{M_{\left\{P\right\}}}}{x_{M_{\left\{P\right\}}}}\right)\\ H_{\left\{P\right\}} & =\left({M_{\left\{P\right\}}}^{T}\;\cdot \;{w_{H}}_{\left\{P\right\}}\right){w_{H}}_{\left\{P\right\}} \end{array} \end{array}\right.$$

If ${M_{\left\{P\right\}}}^{T}\;\cdot \;{w_{H}}_{\left\{P\right\}}<0$ then the projected point is in the lower part of the cone. Working with the upper part of the cone, when this condition is true, the nearest point is the vertex of the $O_{P}$ cone (example in Figure 6). The orthogonal distance *h* is therefore (Equation (Equation 4)):(4)$$h=\left\{\begin{array}{c} \begin{array}{ccc} \sqrt{{\left(M_{\left\{P\right\}}-H_{\left\{P\right\}}\right)}^{T}\left(M_{\left\{P\right\}}-H_{\left\{P\right\}}\right)} & \; & \;\mathrm{si}\;{M_{\left\{P\right\}}}^{T}{w_{H}}_{\left\{P\right\}}⩾0\\ \sqrt{{M_{\left\{P\right\}}}^{T}M_{\left\{P\right\}}} & \; & \;\mathrm{si}\;{M_{\left\{P\right\}}}^{T}{w_{H}}_{\left\{P\right\}}<0 \end{array} \end{array}\right.$$

#### 3.1.3. Quadratic Form of the Cone

To obtain this relationship, one must first note that if the angle between the vectors $\left(X-O_{P}\right)$ and d or $\left(X-O_{P}\right)$ and −d is equal to the angle α, then the point X belongs to the cone. This is equivalent to the following scalar product:$${\left(X-O_{P}\right)}^{T}d=\pm ∥X-O_{P}∥\cos\left(\alpha \right)$$
When squared, the expression becomes:$$\begin{array}{cc} \; & {\left({\left(X-O_{P}\right)}^{T}d\right)}^{2}={\left(∥X-O_{P}∥\cos\left(\alpha \right)\right)}^{2}\\ \Leftrightarrow & \left({\left(X-O_{P}\right)}^{T}d\right)\left({\left(X-O_{P}\right)}^{T}d\right)={\left(X-O_{P}\right)}^{T}\left(X-O_{P}\right)\cos^{2}\left(\alpha \right)\\ \Leftrightarrow & {\left(X-O_{P}\right)}^{T}dd^{T}\left(X-O_{P}\right)-{\left(X-O_{P}\right)}^{T}\left(X-O_{P}\right)\cos^{2}\left(\alpha \right)=0\\ \Leftrightarrow & {\left(X-O_{P}\right)}^{T}\underset{Q}{\underset{︸}{\left[\begin{array}{c} dd^{T}-\cos^{2}\left(\alpha \right)I_{3} \end{array}\right]}}\left(X-O_{P}\right)=0 \end{array}$$
Thus, one can obtain a quadratic function of the cone in the general case (Equation Equation 5), directly showing the parameters d, α and $O_{P}$ which define it:(5)$$Q\left(X\right)={\left(X-O_{P}\right)}^{T}Q\left(X-O_{P}\right)$$

This gives the relationships between the point X and the cone (Equations (6a)–(6f)): (6a)•  IfQ(X)=0thenXbelongstothecone(6b)•  IfQ(X)<0thenXisinsidethecone(6c)•IfQ(X)>0thenXisoutsidethecone(6d)•IfQ(X)=0and(X−OP)Td>0thenXbelongstotheupperpartofthecone(6e)•IfQ(X)<0and(X−OP)Td>0thenXisinsidetheupperpartofthecone(6f)•IfQ(X)>0or(X−OP)Td<0thenXisoutsidetheupperpartofthecone.

### 3.2. Projector Calibration

Projector calibration is an essential step, since it involves estimating the parameters of the cone used in this method and in the Equations (Equation 1) to (6f). As a reminder, the parameters of the cone to be estimated are its vertex $O_{P}$, its direction vector d and its half-angle of aperture α.

Since $O_{P}$ and d represent the relative pose of the cone with respect to the camera, the projector must be fixed with respect to the camera during calibration and during any experimentation. Furthermore, if the projector has a variable half-angle aperture, it is also necessary to ensure that this is fixed.

#### 3.2.1. Three-Dimensional Point Generation for Cone Estimation

The first step in the calibration method is to generate a cloud of 3D points belonging to the cone. To achieve it, the chosen method is to capture several light projection images on a flat surface.

This light projection is the result of the intersection between the projector cone and the flat surface, which is by definition a conic. In this case, it will be an ellipse because one wants a closed shape. The projection of this ellipse into the image is also an ellipse.

If one can estimate the relative pose of this surface with respect to the camera and extract the light contour in the image, which is an ellipse, then one can obtain the real ellipse from the ellipse in the image (Figure 7).

To estimate the relative pose, a chessboard pattern is attached to the surface to estimate its relative pose. This relative pose is obtained by estimating the homography between the calibration pattern and the image plane [43]. As the camera is calibrated, rotation and translation can be estimated from the homography. To extract the ellipse from the image, a contour detection method is used, presented in [44,45]. Then, from the contour points obtained, the best ellipse is estimated using the method of [46], which is based on least squares optimisation.

It is therefore possible to obtain *n* elliptical sections of the cone from *n* images taken at different distances from the surface. For each ellipse extracted in an image, the ellipse in the surface is obtained via the estimated homography between the image plane and this surface. The estimated relative pose allows one to obtain the elliptical section, i.e., the set of 3D points expressed in the camera frame of reference belonging to the ellipse of the surface.

Figure 8 shows an example where the camera has captured three images of an ellipse surrounding a chessboard pattern on the flat surface in three different poses. This is equivalent to obtaining three elliptical sections of the same cone.

Consequently, if *n* elliptical sections are obtained where *m* 3D points are extracted per section, one can generate a cloud of n×m 3D points belonging to the cone.

#### 3.2.2. Cone Estimation

The aim is now to estimate the cone that best approximates the 3D point cloud. This is performed by geometric fitting, i.e., by minimizing the orthogonal distances between the 3D points and the cone.

Let p be the vector containing the six parameters of the cone, which are:the three coordinates of its vertex $O_{P}$;the two angles yaw and pitch in ZYX-Euler convention of its direction vector d, the angle roll not being necessary since a cone has an axis of symmetry;its opening half-angle α.

To estimate the p vector, the Levenberg–Marquardt (LM) iterative method [47] is used. For the first iteration, one needs to choose an initial parameter vector $p_{0}$. This could be the null vector, for example. In practice, measurements will be used, taken with a caliper for the various parameters of $p_{0}$.

At each iteration *i*, LM receives a solution vector $p_{i}$ and the function for calculating the vector of orthogonal distances to be minimized. This function first calculates the rotation and translation of the cone relative to the camera from $O_{P}$ and the yaw and pitch angles of the vector $p_{i}$. It then expresses the 3D points in the cone reference frame and calculates the orthogonal distance vector between the 3D points and the cone using the Equation (Equation 4).

So from $p_{i}$ and this function, LM returns a new solution vector $p_{i+1}$.

When the difference between the sum of the squared orthogonal distances obtained with $p_{j}$ and the calculus obtained with $p_{j+1}$ is less than an arbitrary threshold, this means that LM has reached convergence at iteration *j*. The best solution in the least squares sense is therefore the vector $p_{j}$.

It is important to note that there are an infinite number of cones that share the same elliptical section. On the other hand, there is only one cone that passes through an elliptical section and through a point outside that section. Consequently, this minimization can only work if one has at least two elliptical sections.

### 3.3. Intersection with a Ray

Now that the method to determine the cone parameters using the calibration method is known, the next step is to be able to calculate the intersections between a camera ray and the cone; we are only interested in the intersections with the top of the cone.

A ray is a half-line that can be expressed parametrically using the following Equation (Equation 7):(7)X(t)=O+tu
where O is the starting point of the ray, u is its unit direction vector and *t* is a positive real number such that X(0)=O. Note that it only deals with cases where the point O is outside the cone. This is because the starting point of the camera rays is the optical center in this situation. The point O can nevertheless be inside the lower part of the cone.

If X(t) is an intersection point, then it belongs to the cone. Therefore, to determine the intersections, one can simply calculate the values of *t* which cancel the general polynomial of the cone (Equation (Equation 5)) after replacing X by the expression for X(t): $$\begin{array}{c} \begin{array}{cc} \; & Q\left(X\left(t\right)\right)={\left(X\left(t\right)-O_{P}\right)}^{T}Q\left(X\left(t\right)-O_{P}\right)=0\\ \; & \Leftrightarrow {\left(O+tu-O_{P}\right)}^{T}Q\left(O+tu-O_{P}\right)=0\\ \; & \Leftrightarrow t^{2}\left(u^{T}Qu\right)+t\left(2u^{T}Q\left(O-O_{P}\right)\right)+{\left(O-O_{P}\right)}^{T}Q\left(O-O_{P}\right)=0\\ \; & \Leftrightarrow \;at^{2}+bt+c=0\;\mathrm{with}\;\left\{\begin{array}{c} \begin{array}{cc} a & =\;u^{T}Qu\\ b & =\;2u^{T}Q\left(O-O_{P}\right)\\ c & =\;{\left(O-O_{P}\right)}^{T}Q\left(O-O_{P}\right)=Q\left(O\right) \end{array} \end{array}\right. \end{array} \end{array}$$

### 3.4. Projection of the Generatrices of the Cone in the Image

To understand the relationship between a point on the cone and its projection into the image, one needs to look at the projections of the generatrices of the cone. Since the cone represents the projector, only the upper half of the cone is considered, which means that the generatrices can be limited to the half-lines that start at the vertex $O_{P}$. Note that the study is limited to the case where the cone is oriented in the same direction as the camera, as required by this method. Geometrically, this condition is satisfied if the angle called μ between $z_{C}$ and $d=z_{P}$ is in the interval ]$\frac{\pi }{2}-\alpha ;-\frac{\pi }{2}+\alpha$[ (see Figure 9). In practice, for the camera to be able to see the light projection, the angle μ will be close to 0.

Calculating the projections of the generatrices of the cone

Let us call $\Pi_{g}$ the plane which passes through the optical center $O_{C}$ and through at least one generatrix of the cone. The intersection of this plane with the image plane is the line $\ell_{g}$.

Let $X^{\prime }$ be a point in the image such that $X^{\prime }=KC_{\left\{u\right\}}$, where u is the direction vector of the camera ray associated with $X^{\prime }$. If $X^{\prime }$ is the projection into the image of a 3D point belonging to a generatrix resulting from the intersection between the cone and the plane $\Pi_{g}$, then $X^{\prime }$ belongs to the line $\ell_{g}$. However, if $X^{\prime }$ belongs to the line $\ell_{g}$, it is not necessarily the projection of a point on the cone. To obtain the equation of $\ell_{g}$, the simplest way is to select two points belonging to one of the generatrices (or the generatrix) through which the $\Pi_{g}$ passes, then project them into the image to find the parameters of the line. The same result would be obtained by projecting two points of the plane $\Pi_{g}$ which do not belong to the same camera radius. However, knowing the line $\ell_{g}$ is not enough to understand the projection of a generatrix.

Observation when $\Pi_{g}$ is not tangent to the cone

When the plane $\Pi_{g}$ is not tangent to the cone, it passes through two generatrices $g_{1}$ and $g_{2}$, with $g_{1}$ chosen as the generatrix closest to the optical center $O_{C}$ (see Figure 10).

The direction vectors of these two generatrices are $w_{1}$ and $w_{2}$ such that:$${w_{1}}_{\left\{P\right\}}=\left[\begin{array}{c} \sin\left(\alpha \right)\cos\left(\theta_{1}\right)\\ \sin\left(\alpha \right)\sin\left(\theta_{1}\right)\\ \cos(\alpha ) \end{array}\right]\;\mathrm{et}\;{w_{2}}_{\left\{P\right\}}=\left[\begin{array}{c} \sin\left(\alpha \right)\cos\left(\theta_{2}\right)\\ \sin\left(\alpha \right)\sin\left(\theta_{2}\right)\\ \cos(\alpha ) \end{array}\right]$$
where $\theta_{1}$ and $\theta_{2}$ are the angle between the vector $w_{1}$ and $w_{2}$, respectively, projected in the plane $x_{P}y_{P}$ and the vector $x_{P}$ (see Figure 10). As the plane $\Pi_{g}$ is chosen here as not tangent to the cone, one has $\theta_{1}\neq \theta_{2}$. In this configuration, if $X^{\prime }$ is a point on the line $\ell_{g}$ then it can be both the projection of a point $X_{1}$ belonging to $g_{1}$ and of a point $X_{2}$ belonging to $g_{2}$. The generatrix $g_{1}$ therefore contains the first intersections of the camera ray with the cone, while $g_{2}$ contains the second intersections.

Calculating the two special generatrices when $\Pi_{g}$ is tangent to the cone

There are only two orientations for which the plane $\Pi_{g}$ is tangent to the cone. When this happens, then $\theta_{1}=\theta_{2}$ and the generatrices $g_{1}$ and $g_{2}$ are superimposed. The two possible cases are illustrated in Figure 11. The two angles $\theta_{A}$ and $\theta_{B}$ define the two generatrices $g_{A}$ and $g_{B}$ through which pass the two planes tangent to the cone $\Pi_{g_{A}}$ and $\Pi_{g_{B}}$. It is important to note that the angles $\theta_{A}$ and $\theta_{B}$ depend solely on the parameters of the cone and its pose relative to the camera $T_{\left\{E\to C\right\}}$, which means that they are unique and fixed for a given configuration.

To determine these two angles, a plane tangent to the cone is used, called $\Pi_{T}$, which passes through a generatrix g whose direction vector is w defined by an angle θ. It is represented in Figure 10 passing through the generatrix $g_{2}$ purely for reasons of visibility. The orientation of the plane $\Pi_{T}$ is characterized by the unit normal vector $n_{T}$ such that:(8)$${n_{T}}_{\left\{P\right\}}=\left[\begin{array}{c} \cos(\alpha )\cos(\theta )\\ \cos(\alpha )\sin(\theta )\\ -\sin(\alpha ) \end{array}\right]$$
It is now possible to look for the two solutions of θ so that the plane $\Pi_{T}$ is superimposed on the plane $\Pi_{g_{A}}$ or $\Pi_{g_{B}}$. This superposition only occurs when the plane $\Pi_{T}$ passes through the optical center $O_{C}$ and therefore when the vector $n_{T}$ is orthogonal to the vector ($O_{C}$ − $O_{P}$). This leads us to look for θ when (Equation (Equation 9)):(9)$${{O_{C}}_{\left\{P\right\}}}^{T}{n_{T}}_{\left\{P\right\}}=0\Leftrightarrow \left[\begin{array}{ccc} x_{{O_{C}}_{\left\{P\right\}}} & y_{{O_{C}}_{\left\{P\right\}}} & z_{{O_{C}}_{\left\{P\right\}}} \end{array}\right]\left[\begin{array}{c} \cos(\alpha )\cos(\theta )\\ \cos(\alpha )\sin(\theta )\\ -\sin(\alpha ) \end{array}\right]=0\;\Leftrightarrow x_{{O_{C}}_{\left\{P\right\}}}\cos\left(\alpha \right)\cos\left(\theta \right)+y_{{O_{C}}_{\left\{P\right\}}}\cos\left(\alpha \right)\sin\left(\theta \right)-z_{{O_{C}}_{\left\{P\right\}}}\sin\left(\alpha \right)=0$$
where ${O_{C}}_{\left\{P\right\}}$ is the optical center expressed in the reference frame of the P projector. To solve the Equation (Equation 9), one can pose (Equation (Equation 10)):(10)$$s=\cos\left(\alpha \right)\left[\begin{array}{c} x_{{O_{C}}_{\left\{P\right\}}}\\ y_{{O_{C}}_{\left\{P\right\}}} \end{array}\right]\;\mathrm{and}\;t=\left[\begin{array}{c} \cos(\theta )\\ \sin(\theta ) \end{array}\right]$$

One can rewrite Equation (Equation 9) using the scalar product between the vectors s and t and the angles defined in Figure 12:$$s^{T}t=\overset{∥s∥∥t∥=∥s∥}{\overset{︷}{\cos\left(\alpha \right)\sqrt{x_{{O_{C}}_{\left\{P\right\}}}^{2}+y_{{O_{C}}_{\left\{P\right\}}}^{2}}}}\cos\left(\gamma \right)=z_{{O_{C}}_{\left\{P\right\}}}\sin\left(\alpha \right)$$
The angle γ between the vectors s and t is therefore:$$\gamma =\pm \arccos\left(\frac{z_{{O_{C}}_{\left\{P\right\}}}\sin\left(\alpha \right)}{\cos\left(\alpha \right)\sqrt{x_{{O_{C}}_{\left\{P\right\}}}^{2}+y_{{O_{C}}_{\left\{P\right\}}}^{2}}}\right)=\pm \arccos\left(\frac{z_{{O_{C}}_{\left\{P\right\}}}}{\sqrt{x_{{O_{C}}_{\left\{P\right\}}}^{2}+y_{{O_{C}}_{\left\{P\right\}}}^{2}}}\tan\left(\alpha \right)\right)$$
As for the angle β which defines the orientation of the vector s, its expression is:$$\beta =\mathrm{atan}2(y_{{O_{C}}_{\left\{P\right\}}},x_{{O_{C}}_{\left\{P\right\}}})$$
The solution to Equation (Equation 9) is therefore (Equation (Equation 11)):(11)$$\theta_{A,B}=\beta \pm \gamma =\mathrm{atan}2\left(y_{{O_{C}}_{\left\{P\right\}}},x_{{O_{C}}_{\left\{P\right\}}}\right)\pm \arccos\left(\frac{z_{{O_{C}}_{\left\{P\right\}}}}{\sqrt{x_{{O_{C}}_{\left\{P\right\}}}^{2}+y_{{O_{C}}_{\left\{P\right\}}}^{2}}}\tan\left(\alpha \right)\right)$$
Note that the two angles $\theta_{A}$ and $\theta_{B}$ only depend on the half-angle of the opening α of the cone and the translation between the cone and the camera (translation expressed in the equation by ${O_{C}}_{\left\{P\right\}}$). From the expression of these two angles, one can obtain the expression of the two unit direction vectors $w_{A}$ and $w_{B}$ of the two special generatrices $g_{A}$ and $g_{B}$. The projections of these two generatrices belong to the lines $\ell_{g_{A}}$ and $\ell_{g_{B}}$ which are the intersections of the image plane with the planes $\Pi_{g_{A}}$ and $\Pi_{g_{B}}$.

Splitting the cone into two areas using the two special generatrices

Using the generatrices $g_{A}$ and $g_{B}$, one can divide the cone into two surfaces as shown in Figure 13. The cyan surface named $S_{1}$ (and the magenta surface named $S_{2}$, respectively) contains the set of generatrices containing all the first (and second, respectively) possible intersections between a camera beam and the cone. These generatrices were previously named $g_{1}$ (and $g_{2}$, respectively).

In the image plane, this shows that one can delimit the areas where the camera rays can intersect the cone. The cyan area, surrounded by the ellipse $E_{\infty }^{\prime }$, contains the projections of part of the set of 3D points belonging to the surface $S_{1}$. The magenta area contains the projections of the other part of the set of 3D points belonging to the surface $S_{1}$ and contains the projections of the set of 3D points belonging to the surface $S_{2}$. These areas are bounded by:the lines $\ell_{g_{A}}$ and $\ell_{g_{B}}$;the ellipse $E_{\infty }^{\prime }$ which is the set of vanishing points of the generatrices. These points define a first bound on all the projections of the generatrices;the projection of the vertex $O_{P}$ into the image plane, named $O_{P}^{\prime }$. It defines a second bound on all the projections of the generatrices.

In the configuration illustrated in Figure 13, each generatrix projection is therefore a segment whose limits are the vanishing point of the generatrix belonging to the ellipse $E_{\infty }^{\prime }$ and the point $O_{P}^{\prime }$.

However, there are other configurations where the projections of the generatrices onto the image plane are not segments but lines. Indeed, if $z_{{O_{P}}_{\left\{C\right\}}}=0$, then the line passing through $O_{P}$ and $O_{C}$ is parallel to the plane, which implies that the projection of the vertex $O_{P}$ into the image plane is a point at infinity (or ideal point) and that the lines $\ell_{g_{A}}$ and $\ell_{g_{B}}$ are parallel.

The case where $z_{{O_{P}}_{\left\{C\right\}}}<0$ is illustrated in Figure 14 which shows the lower part of the cone behind the camera. This part is shown in brown and its projection in the image plane is also in brown. This projection is obviously fictitious, as this part cannot be seen by the camera even if it had an infinite field of view. The point $O_{P}^{\prime }$, also fictitious, is still the intersection of the lines $\ell_{g_{A}}$ and $\ell_{g_{B}}$, but this time it is on the other side in comparison to that shown in Figure 13.

As for the magenta and cyan areas in the image plane, they are still delimited by the lines $\ell_{g_{A}}$ and $\ell_{g_{B}}$ and the ellipse $E_{\infty }^{\prime }$ but extend towards infinity on the side where the lines $\ell_{g_{A}}$ and $\ell_{g_{B}}$ diverge.

Therefore, if $z_{{O_{P}}_{\left\{C\right\}}}\leq 0$, then each generatrix projection is a half-line whose only boundary is its vanishing point belonging to the ellipse $E_{\infty }^{\prime }$. However, working with real cameras and not with the image plane, which is infinite, each generatrix will have a segment as its projection into the image.

This division of the cone into two surfaces leads to several results:Any ray associated with a point in the cyan area has its unique intersection with the cone a 3D point belonging to the surface $S_{1}$.Any ray associated with a point in the magenta area has two intersections with the cone, the first of which belongs to the surface $S_{1}$ and the second to the surface $S_{2}$. These two intersections are superimposed if the point in the magenta area belongs to the line $\ell_{g_{A}}$ or to the line $\ell_{g_{B}}$.Any ray associated with a point outside the magenta area and the cyan area does not have any intersection with the cone.

### 3.5. Intersection Selection and Triangulation

Now, everything is ready to develop a method for determining which intersection to choose when the ray associated with a pixel in the contour has two intersections with the cone. To better understand the various stages of the procedure, let us take as an example the 3D contour of light whose associated 3D curve is circular and results from the intersection between a plane and the cone.

The example of the circular contour is illustrated in Figure 13 which contains the following elements:$C^{\prime }$ is the closed 2D curve corresponding to the contour in the image with $X^{\prime }$ a 2D point on the curve.C is our circular 3D curve corresponding to the contour of the light in the scene, where X is a 3D point on the curve which has $X^{\prime }$ as its projection. The point X therefore corresponds to the correct intersection between the cone and the camera ray associated with $X^{\prime }$.The areas containing the first and second intersections with the cone are delimited by the two generatrices $g_{A}$ and $g_{B}$ which divide the cone into two surfaces $S_{1}$ and $S_{2}$. They also divide C into two curves, one in cyan called $C_{1}$ and the other in magenta called $C_{2}$.A and B are the two intersections of $g_{A}$ and $g_{B}$, respectively, with the curve C and are thus the only two 3D points common to the curves $C_{1}$ and $C_{2}$.The cyan curve named ${C^{\prime }}_{1}$ and the magenta curve named ${C^{\prime }}_{2}$ are the projections of the curves $C_{1}$ and $C_{2}$, respectively.$A^{\prime }$ and $B^{\prime }$ are the projections of A and B and are therefore the only two points common to the curves ${C^{\prime }}_{1}$ and ${C^{\prime }}_{2}$. They belong to both the curve $C^{\prime }$ and the lines $\ell_{g_{A}}$ and $\ell_{g_{B}}$. So, in theory, they correspond to the two unique points of tangency of the lines $\ell_{g_{A}}$ and $\ell_{g_{B}}$ with $C^{\prime }$.

Using the points of tangency, $A^{\prime }$ and $B^{\prime }$, will allow one to divide the 2D curve $C^{\prime }$ in two via the line $\left(A^{\prime }B^{\prime }\right)$. However, in practice:the curve $C^{\prime }$ is discrete,the light contours are extracted in a perfectible way,the camera calibration and the cone estimation have uncertainties, so the lines $\ell_{g_{A}}$ and $\ell_{g_{B}}$ also have uncertainties.

All this means that the lines $\ell_{g_{A}}$ and $\ell_{g_{B}}$ are not exactly tangent to the curve $C^{\prime }$. The method for determining the points $A^{\prime }$ and $B^{\prime }$ must therefore be adapted to this constraint.

Once these two points have been obtained and the curve $C^{\prime }$ has been divided in two, it remains to determine which of the two curves is ${C^{\prime }}_{1}$ or ${C^{\prime }}_{2}$. To do this, one needs to look at the relative position of the vertex $O_{P}$ in relation to the camera. Figure 15 shows that the vertex $O_{P}$ and the curve ${C^{\prime }}_{2}$ lie in the same half-space bounded by the plane passing through $O_{C}$ and the line $\left(A^{\prime }B^{\prime }\right)$. This property is always true whatever the pose of the cone.

In the image plane, this property can be expressed by defining a point $p_{\mathrm{ref}}$ such that $p_{\mathrm{ref}}$ and the curve ${C^{\prime }}_{2}$ lie in the same half-plane bounded by the line $\left(A^{\prime }B^{\prime }\right)$.

There are an infinite number of solutions for $p_{\mathrm{ref}}$ in the image plane which satisfy this condition. Therefore $p_{\mathrm{ref}}$ is forced to belong to the circumscribed circle of the rectangle delimiting the image, as can be seen in Figure 15. The segment between $p_{\mathrm{ref}}$ and the center of the image forms an angle with the horizontal axis of the image called ν. This gives (Equation (Equation 12)):(12)$$p_{\mathrm{ref}}=\sqrt{w^{2}+h^{2}}\left[\begin{array}{c} \cos(\nu )\\ \sin(\nu ) \end{array}\right]$$
where *w* and *h* are the width and height of the image in pixels, respectively. For the condition to be true, one possible solution for ν is to indicate the direction of $O_{P}$ relative to the optical center $O_{C}$. To obtain this angle, $O_{P}$ is orthogonally projected onto the plane normal to the camera axis passing through $O_{C}$, which gives the point $H_{P}$ illustrated in Figure 15. This gives ν which is the angle formed by the vectors $\left(H_{P}-O_{C}\right)$ and $x_{C}$ (Equation (Equation 13)):(13)$$\left(H_{P}-O_{C}\right)=\left[\begin{array}{c} x_{{O_{P}}_{\left\{C\right\}}}\\ y_{{O_{P}}_{\left\{C\right\}}}\\ 0 \end{array}\right]\;\mathrm{donc}\;\nu =\arctan\left(\frac{y_{{O_{P}}_{\left\{C\right\}}}}{x_{{O_{P}}_{\left\{C\right\}}}}\right)$$
Calculating this angle ν and the point $p_{\mathrm{ref}}$ allows one to deduce which of the two curves is ${C^{\prime }}_{2}$.

Once the curves ${C^{\prime }}_{1}$ and ${C^{\prime }}_{2}$ have been obtained, it is then possible to obtain the 3D contour. Indeed, if a point $X^{\prime }$ belongs to ${C^{\prime }}_{1}$, then the first intersection of the camera ray associated with it must be chosen, and if it belongs to ${C^{\prime }}_{2}$, the second must be chosen. This is because any point $X^{\prime }$ of ${C^{\prime }}_{1}$ is the image of a 3D point X of $C_{1}$, and any point $X^{\prime }$ of ${C^{\prime }}_{2}$ is the image of a 3D point X of $C_{2}$.

### 3.6. Test of the Method in Simulation

Contour simulation

To illustrate the different stages of the method, a gallery model will be used. Figure 16 shows the camera represented by a pyramid and the projector represented by a cone positioned inside the model. In this example, the axis of the cone is parallel to the axis of the camera. The vertex $O_{P}$ is positioned above and to the right of the camera; it belongs to the plane which passes through the optical center $O_{C}$ and which is orthogonal to the camera axis. Consequently, the projection of $O_{P}$ in the image plane is a point at infinity (an ideal point) and the projections of the generatrices all belong to lines parallel to each other.

The 3D curve C is the intersection between the cone and the model. The set of 3D points on this curve is thus obtained from the intersections of the generatrices of the cone with the model, and they are shown in black in Figure 16. The number of 3D points is equal to the number of generatrices chosen to represent the cone. These 3D points are then projected onto the image to obtain a set of 2D points belonging to the curve curveProjCone. These 2D points are shown in black in Figure 16.

The idea now is to obtain the 3D points of the curve C from the 2D points of the curve $C^{\prime }$. Assume that the parameters of the cone and the camera are known.

Calculating the generatrices and the lines containing their projection

The first step is to calculate the direction vectors $w_{A}$ and $w_{B}$ of the generatrices $g_{A}$ and $g_{B}$. To do this, one needs to calculate the angles $\theta_{A}$ and $\theta_{B}$ using the Equation (Equation 11).

Once the direction vectors have been obtained, two 3D points are selected on $g_{A}$ and $g_{B}$, projected into the image, and the parameters of the lines $\ell_{g_{A}}$ and $\ell_{g_{B}}$ are computed. The two lines are defined by their respective unit direction vectors $u_{A}$ and $u_{B}$ and by $A_{1}^{\prime }$ and $B_{1}^{\prime }$, two points belonging to the two lines, respectively.

Determining the points $A^{\prime }$ and $B^{\prime }$

Now, one needs to obtain the points $A^{\prime }$ and $B^{\prime }$ which are theoretically the points of tangency of the lines $\ell_{g_{A}}$ and $\ell_{g_{B}}$ with the curve $C^{\prime }$. The approach will be only presented for the point $A^{\prime }$ since it will be the same for $B^{\prime }$.

A first solution would be to say that $A^{\prime }$ is the point belonging to the line $\ell_{g_{A}}$ closest to the curve $C^{\prime }$. However, a slightly more generic approach has been chosen, the result of which is illustrated in Figure 17. It consists of considering $A^{\prime }$ as the point belonging to the line $\ell_{g_{A}}$ which minimizes the sum of the squared distances between itself and the x% of the points on the curve $C^{\prime }$ closest to the line $\ell_{g_{A}}$. Let *t* be the parameter of the line. Therefore, looking for *t* such that (Equation (Equation 14)):(14)$$t=\underset{t}{\mathrm{argmin}}\left\{\sum_{i=1}^{n}\frac{{\left(A^{\prime }-X_{i}^{\prime }\right)}^{T}\left(A^{\prime }-X_{i}^{\prime }\right)}{h(X_{i}^{\prime })}\right\}\;\mathrm{with}\;A^{\prime }=A_{1}^{\prime }+tu_{A}$$
where:$X_{i}^{\prime }$ are the x% of the points on the curve $C^{\prime }$ closest to the line $\ell_{g_{A}}$.$h:R^{3}\to R^{+}$ is a function whose aim is to reduce the impact of the points $X_{i}^{\prime }$ far from the line $\ell_{g_{A}}$ in minimization.

Figure 17 shows the points $A^{\prime }$ and $B^{\prime }$ obtained, with 40% of the points closest to $\ell_{g_{A}}$ in yellow and 40% of the points closest to $\ell_{g_{B}}$ in orange. Developing the sum to be minimized in Equation (Equation 14) gives:$$\begin{array}{cc} \sum_{i=1}^{n}\frac{{\left(A^{\prime }-X_{i}^{\prime }\right)}^{T}\left(A^{\prime }-X_{i}^{\prime }\right)}{h(X_{i}^{\prime })} & =\sum_{i=1}^{n}\frac{{\left(A_{1}^{\prime }+tu_{A}-X_{i}^{\prime }\right)}^{T}\left(A_{1}^{\prime }+tu_{A}-X_{i}^{\prime }\right)}{h(X_{i}^{\prime })}\\ \; & =\sum_{i=1}^{n}\frac{{\left(tu_{A}+v_{i}\right)}^{T}\left(tu_{A}+v_{i}\right)}{h(X_{i}^{\prime })}\;\mathrm{with}\;v_{i}=A_{1}^{\prime }-X_{i}^{\prime }\\ \; & =\sum_{i=1}^{n}\frac{\left(t^{2}u_{A}^{T}u_{A}+2tu_{A}^{T}v_{i}+v_{i}^{T}v_{i}\right)}{h(X_{i}^{\prime })}\\ \; & =t^{2}u_{A}^{T}u_{A}\sum_{i=1}^{n}\frac{1}{h(X_{i}^{\prime })}+t2\sum_{i=1}^{n}\frac{u_{A}^{T}v_{i}}{h(X_{i}^{\prime })}+\sum_{i=1}^{n}\frac{v_{i}^{T}v_{i}}{h(X_{i}^{\prime })}\\ \; & =at^{2}+2bt+c \end{array}$$
with
$$\left\{\begin{array}{c} \begin{array}{c} a=u_{A}^{T}u_{A}\sum_{i=1}^{n}\frac{1}{h(X_{i}^{\prime })}\\ b=\sum_{i=1}^{n}\frac{u_{A}^{T}v_{i}}{h(X_{i}^{\prime })}\\ c=\sum_{i=1}^{n}\frac{v_{i}^{T}v_{i}}{h(X_{i}^{\prime })} \end{array} \end{array}\right.$$
Since *a* must be positive, the polynomial $at^{2}+2bt+c$ has a minimum when its derivative cancels, i.e., when:$$2at+2b=0\Rightarrow t=-\frac{b}{a}$$
This gives:$$A^{\prime }=A_{1}^{\prime }-\frac{b}{a}u_{A}$$
Using the same procedure, the point $B^{\prime }$ can be obtained. This produces the line $\left(A^{\prime }B^{\prime }\right)$ (Figure 17) which will be used to divide the curve $C^{\prime }$ in two.

Separation of the curve $C^{\prime }$ for intersection selection

The points $A^{\prime }$ and $B^{\prime }$ are determined in a perfectible way in reality. Therefore, a minimum distance is defined around the line $\left(A^{\prime }B^{\prime }\right)$ called *e* which the points of the curve $C^{\prime }$ must respect. Points whose distance orthogonal to the line is less than *e* are considered indeterminate and will therefore not be taken into account in the 3D reconstruction of the contour. In practice, the value of *e* will depend on the quality of the measurements taken.

This separation of the $C^{\prime }$ curve into two curves is illustrated in Figure 18 with ${C^{\prime }}_{1}$ in cyan and ${C^{\prime }}_{2}$ in magenta.

To obtain ${C^{\prime }}_{1}$ or ${C^{\prime }}_{2}$, the point $p_{\mathrm{ref}}$ is used (Equation (Equation 12)) obtained via the angle ν (Equation (Equation 13)) and by applying the following two conditions:X′∈C′2ifsignnA′B′T(X′−A′)=−signnA′B′T(pref−A′)X′∈C′1ifsignnA′B′T(X′−A′)=−signnA′B′T(pref−A′)
with $n_{A^{\prime }B^{\prime }}$, the normal vector to the line $\left(A^{\prime }B^{\prime }\right)$.

Three-dimensional reconstruction

Now, points $X^{\prime }$ which are on the curve $C^{\prime }$ are known as they belong to the curve ${C^{\prime }}_{1}$ or the curve ${C^{\prime }}_{2}$. One therefore can apply what was presented in Section 3.3 and Section 3.5 to reconstruct these points in 3D.

Figure 19 takes the 3D scene from Figure 16 and adds the reconstructed 3D points. The calculated differences between these reconstructed 3D points and the initial 3D points of the simulation (obtained via the intersection between the generatrices of the cone and the gallery model) are below $10^{-8}$ m, which is very small. These deviations are due to numerical errors in the calculations at the various stages. This proves that these reconstructed 3D points are the same as the initial 3D points and that our method therefore works. Now the method will be tested with real data.

## 4. Results

The local heritage offers access, between two laboratory buildings, to the Saint-Clément aqueduct (more commonly known as the Arceaux aqueduct) which was built in the 18th century. It is no longer in use, but in dry periods it is an ideal experimental platform. The gallery is accessible via trapdoors, which, once closed, plunge one into total darkness. This confined space recreates experimental conditions similar to those in a karstic environment, but in a dry environment whose dimensions are easy to measure. Indeed, the aqueduct is simply a long corridor whose left and right walls are parallel and separated by a width of 62 cm, which were measured with a laser rangefinder.

### 4.1. Experimental Setup

The camera used is the Nikon D7000 reflex with a 18 mm focal length and a resolution of 2464×1632 pixels (PlongeImage, Bordeaux, France).

The projector is a DIVEPRO M35 (La Palanquee, Palavas, France) dive light with an aperture angle of 145° in air and 90° in water. As the aperture angle is too large to capture the full projection of light in this experiment, a custom-made plastic cylinder (obtained from the 3D printer) was added to reduce it (Figure 20), as can be seen in Figure 21. Its internal diameter is the same as the diameter of the projector, i.e., *d* = 53.8 mm. The cylinder protrudes from the projector by l=93 mm. It is possible to calculate the half-angle of aperture α as a function of the lengths *l* and *d*:$$\alpha =\arctan\left(\frac{d/2}{l}\right)=\arctan\left(\frac{26.9}{93}\right)=16.13{}^{\circ}$$
This value will be used to check the order of magnitude of future α values for the cone estimation.

The camera and projector are placed on a rigid support as shown in Figure 21. They had to be fixed to the support because the slightest rotation of one in relation to the other alters the transformation between the camera and the projector, which is estimated during the projector calibration.

To form an idea of the magnitude of this transformation, a caliper was used to measure the translations between the camera and the projector along the camera’s *x*, *y* and *z* axes. The rotation between the two is supposed to be small because an identical orientation for both was chosen. Here is a summary of the various measurements of the projector parameters (lengths in m):$$\left\{\begin{array}{c} \begin{array}{cc} \alpha & \approx \;\;16.13{}^{\circ}\\ {O_{P}}_{\left\{C\right\}} & \approx \;{\left[\begin{array}{ccc} 0.2 & 0 & -0.07 \end{array}\right]}^{T}\\ ∥{O_{P}}_{\left\{C\right\}}∥ & \approx \;0.212m\\ d_{\left\{C\right\}} & \approx \;{\left[\begin{array}{ccc} 0 & 0 & 1 \end{array}\right]}^{T} \end{array} \end{array}\right.$$

### 4.2. Camera Calibration Results

To calibrate the camera, Zhang’s method [42] was used with the help of eight images of a flat chessboard test pattern measuring 7 × 10 squares and 36 mm wide (Figure 22). In each of the eight images, the position of the test pattern relative to the camera is different (Figure 23), which is a 3D representation of the test pattern in eight different positions. Here are the camera parameters estimated via calibration:the intrinsic matrix *K* obtained is:$$K=\left[\begin{array}{ccc} 1908.56 & 0.0 & 1227.53\\ 0.0 & 1909.94 & 832.48\\ 0.0 & 0.0 & 1.0 \end{array}\right]$$the resulting focal length is 18.2 mm.
It can be seen that the orders of magnitude of the various camera parameters are consistent with the manufacturer’s parameters. In addition, the average reprojection errors are of the order of 1/10 of a pixel. This level of error is quite satisfactory for all eight shots, each containing 56 calibration points.

### 4.3. Projector Calibration Results

To calibrate the projector, i.e., estimate the parameters of the cone, a white wall was used on which a chessboard pattern was hung. The first step is to obtain 3D points belonging to the cone by detecting the contour of the light projected onto the wall, as explained in the Section 3.2. To maximise the contrast of the light projected onto the white wall, the only light source in the room is the projector.

For this calibration, five images were captured of light projected onto the wall with the axis of the cone almost orthogonal to the wall (Figure 24). The shape of the projected light therefore resembles a circle.

Each detected contour is transformed into an ellipse, shown in blue in Figure 24. To obtain the ellipses, the method detailed in [46] was used, but iteratively with the RANSAC algorithm [48] to eliminate certain outlying contour points. The contour points eliminated by RANSAC are shown in red in Figure 24 while the points used to obtain the ellipses are in green.

Using the pattern detected in each image, the equations for the five planes were obtained. From these five planes and the five ellipses obtained, the five elliptical sections are calculated. They are represented in 3D in Figure 25 with the camera and the test pattern.

The image on the left shows all the elements expressed in the chessboard pattern frame, so one can see the different camera positions used to capture the five images.

The image on the right shows all the elements expressed in the camera frame. It is in this frame of reference that the elliptical sections must be expressed in order to estimate the cone.

Three-dimensional points are extracted and expressed in the camera frame of reference for each elliptical section. In this case, 200 points per section were recovered, making a total of 1000 3D points. Note that these 3D points are at a distance from the camera frame of between 1.50 m and 2.17 m.

Now that 1000 3D points have been obtained, which are supposed to belong to the cone, the next step is to define an initial cone for the iterative minimization algorithm. This is shown in red in Figure 26. An initial cone is chosen, that is far from the solution for greater visibility in Figure 26 and also to prove that the method is capable of converging even when the initial cone is far from the solution. Obviously, this is an experimental result and not a mathematical proof. This optimisation problem is probably non-convex; therefore, an initial cone close to the solution should ideally be chosen using the aperture angle previously calculated and the relative pose measurement between the projector and the cone.

The algorithm is then applied to obtain the optimised cone shown in gray. Its estimated parameters (lengths in meters) are:α=14.79°${O_{P}}_{\left\{C\right\}}={\left[\begin{array}{ccc} 0.1956 & 0.0092 & -0.0759 \end{array}\right]}^{T}$$∥{O_{P}}_{\left\{C\right\}}∥$ = 0.210 m$d_{\left\{C\right\}}={\left[\begin{array}{ccc} -0.013 & 0.080 & 0.997 \end{array}\right]}^{T}$

To ensure the validity of the cone obtained, the estimated parameters are compared with the previously measured parameters. Table 1 compares the estimated opening angle and the estimated gap with the measurements. As the measurements can be improved, this comparison at least proves that the order of magnitude of the estimated parameters is consistent.

Another point to check is the minimization error for estimating the cone. In this case, this error is directly linked to the orthogonal distances between the 3D points of the elliptical sections and the cone, since it is the root mean square (RMS) of these distances that the algorithm is trying to minimize during estimation. In Figure 26, the color of each 3D point is defined by its orthogonal distance from the cone via the colormap Jet (the colormap bar is shown in the figure) and the following function:$$\begin{array}{cc} f:R^{+} & \to [0,1]\\ d & \mapsto \frac{d}{d+0.1} \end{array}$$
where *d* is the orthogonal distance, and *m* is chosen as the median of all these distances (here, *m* = 1.4 mm).

The statistics for these distances can be found in Table 2. In particular, a maximum distance of 4.2 mm and an overall mean of 1.5 mm are obtained. The overall RMS is 1.9 mm, which corresponds to approximately 0.13% of 1.50 m which is the smallest distance between a 3D point of an elliptical section and the camera frame. Table 3 represents the statistics between the 3D points of each elliptical section and the reconstructed 3D points. A maximum distance of 226.1 mm and an overall mean of 29.0 mm are obtained.

In view of the analysis of the orthogonal distances, the minimization is satisfactory, confirming once again that the estimated parameters of the cone are at least consistent with their true values.

### 4.4. Three-Dimensional Results

The shape of the aqueduct is a long narrow corridor, which makes it possible to fully distinguish the light projection in the images. Six images were captured, taken at different distances from a chessboard pattern placed in the aqueduct. For each image, the light contours were checked manually, to isolate the triangulation method from the contour detection method, which is still in need of improvement. For each contour extracted, part of the contour points were manually delimited on the left wall and another part on the right wall. These delimitations can be seen in Figure 27. It shows an example of an extracted contour in two images, with contour points on the left wall in red, on the right wall in blue and the rest of the contour points in green.

Only the contours of the left and right walls will be used for triangulation, as this will allow us to check that the 3D points obtained respect the dimensions of the aqueduct corridor. To obtain these 3D points, it is necessary, as usual, to define which intersection is relevant for each radius associated with the contour points. In this situation, one can see that each contour point on the left wall implies a first intersection; additionally, each contour point on the right wall implies a second intersection. Figure 28 illustrates the method presented in Section 3.4 and Section 3.5 for the two images in Figure 27:Lines $\ell_{g_{A}}$ in yellow, $\ell_{g_{B}}$ in orange and their intersection with the curve (Figure 28a,d)Separation of the curves (Figure 28b,e)Intersections with the walls left in cyan and right in magenta (Figure 28c,f)

Figure Figure 29a shows the result of the reconstruction of the contours of the six images. The shape of the 3D reconstructions of the contours is close to a hyperbola, which is consistent since the intersection between a plane and a cone in this configuration is supposed to be a hyperbola.

A comparison of this 3D reconstruction with reality is now possible. The aqueduct is a corridor whose left and right walls are parallel and separated by a width of 62 cm. One can therefore check that:the 3D points resulting from the left part and the right part, respectively, of the contour belong to the same plane so they are supposed to be coplanar,If a plane is estimated from the left 3D points and a plane from the right 3D points, they must be parallel and separated by a distance of approximately 62 cm.

As shown in Figure 29b, estimations are obtained for the left wall in red and for the right wall in blue by using a least square method.

To evaluate coplanarity, the orthogonal distance between 3D points and their associated plane is computed. For left points, Table 4a presents the results for the six positions of the sensor and Table 4b presents the results for all positions with an average deviation of 13.9 mm. For right points, Table 4c presents the results for the six positions of the sensor and Table 4d presents the results for all positions with an average deviation of 24.1 mm.

To check the gap between the two estimated planes, the distances are computed between the left 3D points and the right 3D points, respectively, and the right and left plane, respectively. To estimate the distance between the left points and the right plane, Table 5a presents the results for the six positions of the sensor and Table 5b presents the results for all positions with an average distance of 584.9 mm. To estimate the distance between the left points and the right plane, Table 5c presents the results for the six positions of the sensor and Table 5d presents the results for all positions with an average distance of 583.6 mm. Both distances are near the measured distance of 620 mm.

The two estimated planes are almost parallel since the angle calculated between the two normal vectors is 178.9°.

## 5. Discussion

Method

This original approach is based on the principles of structured light, leveraging the precise projection of light contours onto the karstic surfaces. It functions by capturing the resultant light patterns with a camera leading to reconstruct detailed three-dimensional representations of the subsurface structures. A classical method is used for the camera calibration. However, for the projector calibration estimation, each step is precisely detailed, namely the 3D point generation, the cone estimation, the estimation of the 3D points obtained by the intersection of the cone and its projection in the camera image plane. The main limitation of this approach is the projector contour detection. Indeed, since the boundary between the light and dark zones is not well defined, a detection method based on threshold brightness will be imprecise. It could be solved using a laser cone projector; however, due to the experimental conditions and the fact that the system will be carried by divers, it will be dangerous for their eyes. An active contour model such as that presented in ref. [49] could be used.

Experimental setup

For the validation of the method, a reflex Nikon D7000 with an underwater housing was used, allowing a good resolution (2464 × 1632 pixels), but also allowing for the possibility of adjusting the many settings that can be changed on this type of camera. However, its size remains substantial. The projector (DIVEPRO M35) has a wide angle of aperture and needs a custom-made cylinder to reduce its size. Maybe a dedicated projector with a correct aperture could be better. Also, to explore a new configuration of this system, some simulations were made with one projector and four cameras (Figure 30) to solve the error of the transition zone (see Figure 19). Another idea concerns using several projectors with localization sensors with respect to one or more cameras to have different configurations and measurement redundancy.

An embedded computer would allow the detection and 3D reconstruction directly.

Calibration results

For the camera, the results in dry conditions with a standard chessboard calibration pattern are correct enough, with fewer than 1/10 pixels. However, for underwater calibration, the results are dependent on water turbidity. A solution could be to measure the refraction index of the water and then apply a correction to the camera parameters. For the projector, the same kind of remarks can be made. A first estimation of the cone parameters is performed in dry conditions allowing for an estimation of the errors on its parameters, namely 8% for the angle α and 1% for its position with respect to the camera. Performing the experiment in dry conditions, also knowing the refraction index, could provide useful information to compute the new value of the aperture angle α, avoiding complex experimentation.This calibration is also dependent on the camera calibration; therefore, if the camera calibration is correct, the projector calibration will be accurate.

Three-dimensional reconstruction

The experimentation was conducted in dry conditions with known dimensions of the “cave” (aqueduct). The planes estimation was obtained with a mean value less than 2.5 cm, the distance estimation between the planes was obtained with a mean value less than 3 cm and the angle estimation between the normal range of the planes was obtained with a mean value near 180°. Validation of the method in wet conditions should also be acceptable. However, experimentation must be performed in dry conditions, first in a pool with good visibility conditions. Then a second experiment must be performed in a real cave. One can be reminded that the sites are dependent on administrative authorizations and especially weather conditions; currently, heavy rains make them impassable. Experiments were carried out in a pool a few months ago, but unfortunately the turbidity was such that the detection of the halo contours was problematic, not allowing for a presentation of these results. A new experiment is planned in the coming weeks.

## 6. Conclusions

The aim of this paper was to propose a 3D reconstruction solution designed for underwater galleries found in karstic environments. The study of these environments represents a considerable challenge for the future, as they could provide part of humanity’s water needs. Exploration remains the best approach for acquiring reliable data on the structure, and it is mainly carried out by qualified divers using topography methods with Ariane wires and section-by-section surveys. However, on extensive karstic networks, this approach presents a major risk, as the duration of these surveys is time-consuming, leading divers to long stops. The future is therefore more likely to lie with robotics, but there are still many technical challenges to be overcome before underwater drones that are sufficiently autonomous can be sent out to explore this type of environment. The present research proposed an original method of 3D scanning in this type of environment. This approach uses a combination of a camera and a cone-shaped projector. The parameters of the light cone have been fixed, namely its center, its direction vector and its angular aperture. A method for calibrating the cone was presented, enabling the position of the 3D points obtained by intersecting the visual rays of the contour of the projector halo as seen by the camera with the cone whose parameters have been estimated. The calibration of the camera is performed thanks to a classical approach. A simulation of this method was presented and the results validate the proposal. The first experiment in real conditions was conducted in a disused aqueduct (without water) which had the advantage of reproducing an environment close to that of a karstic aquifer, namely narrow, without light and above all without water to continue the evaluation. The camera calibration was validated with an error reconstruction less than 0.1 pixel and camera parameters close to technical values. The cone calibration led to an angles of error less than 2 degrees and distance with respect to the camera of less than 2 mm. The results were validated by testing the coplanarity of the aqueduct walls to within 25 mm and estimating the known distance between these walls to within 36 mm. Experiments have been performed in real karstic environments; however, due to complex meteorological conditions, visual acquisition was not possible. Thus, future experiments will be performed in a more controlled environment, such as a swimming pool, and then finished in a real aquifer with clearer water.

## Figures and Tables

**Figure 1 sensors-24-04024-f001:**
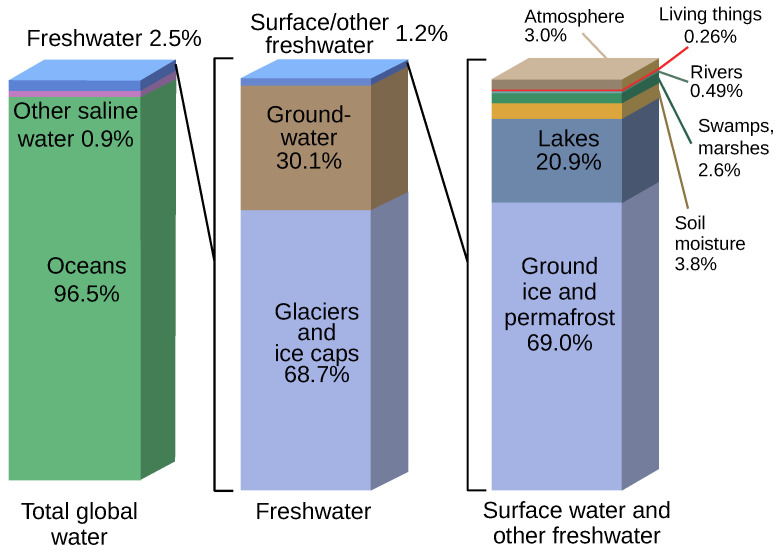
Earth’s water distribution.

**Figure 2 sensors-24-04024-f002:**
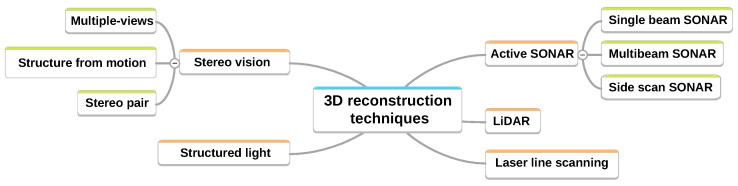
Three-dimensional reconstruction methods in underwater environments.

**Figure 3 sensors-24-04024-f003:**
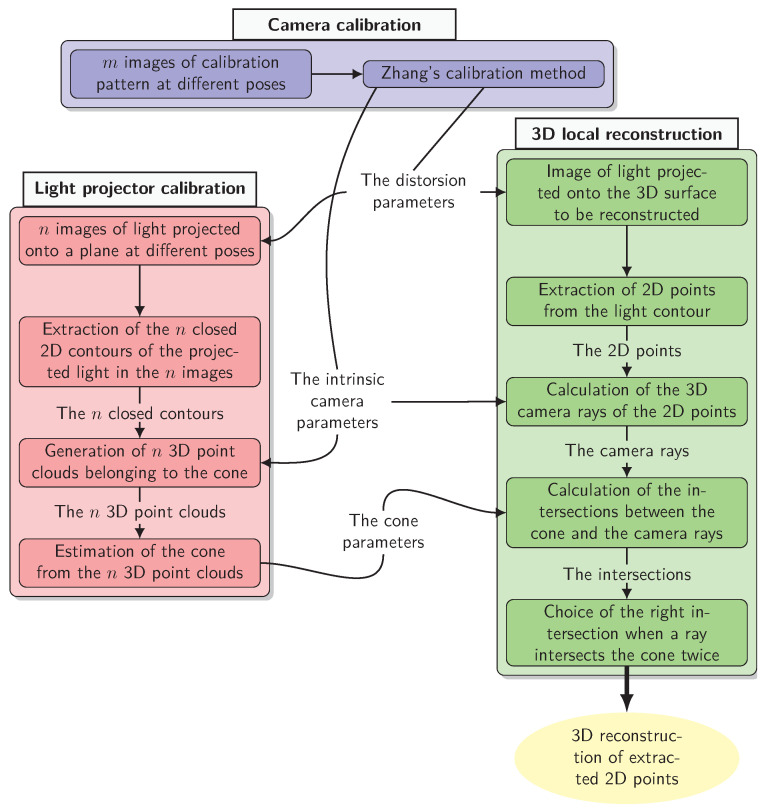
Flowchart of the camera + projector method.

**Figure 4 sensors-24-04024-f004:**
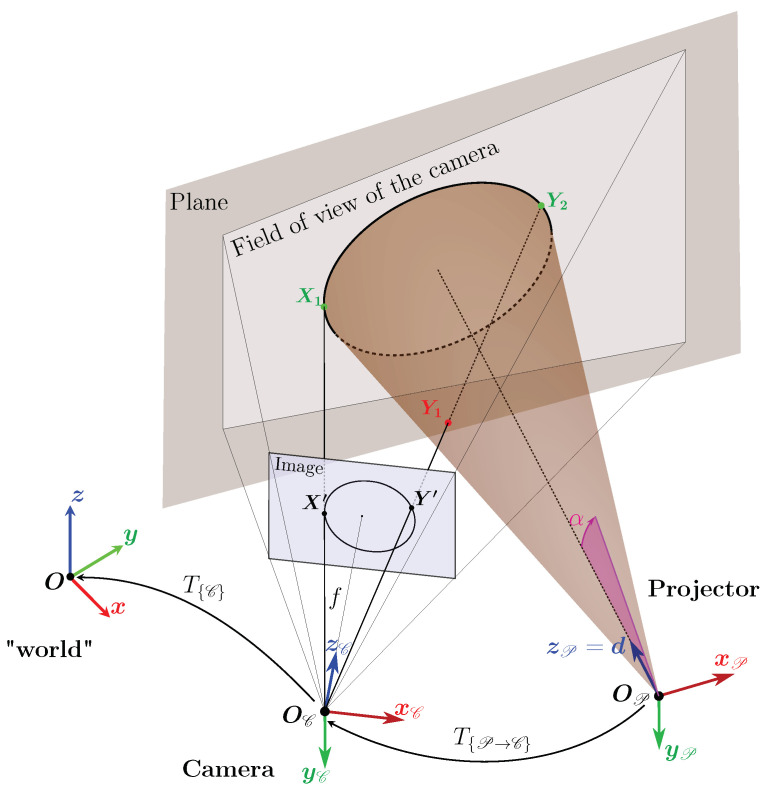
Diagram of the system consisting of the light projector represented by a cone of revolution and the camera observing the light projection on a plane.

**Figure 5 sensors-24-04024-f005:**
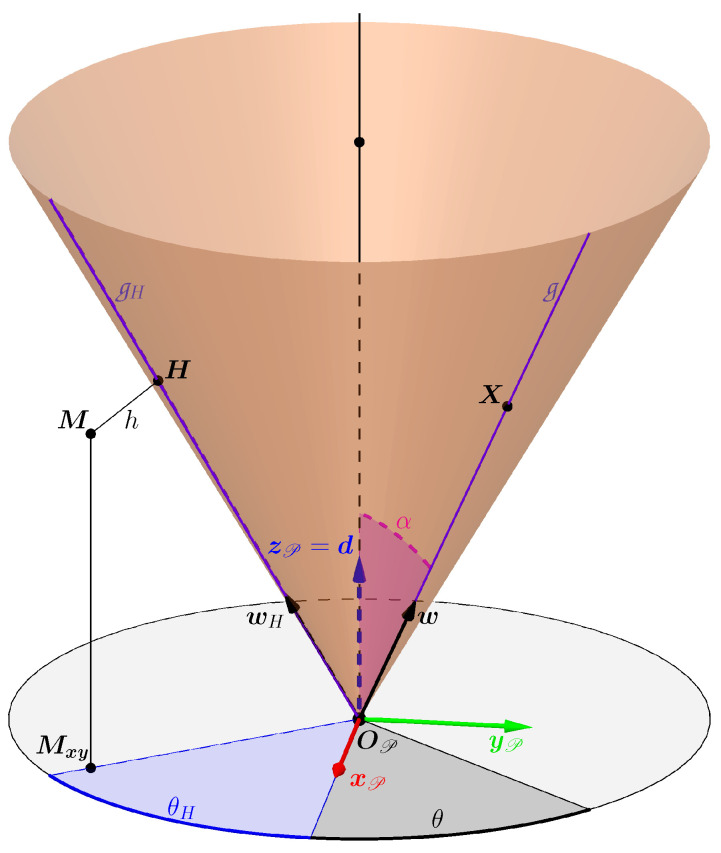
Parameterization of the cone using its generatrices. With this parameterization, one can find the closest generatrix to an external point M and thus find the orthogonal projection H of this point on the cone.

**Figure 6 sensors-24-04024-f006:**
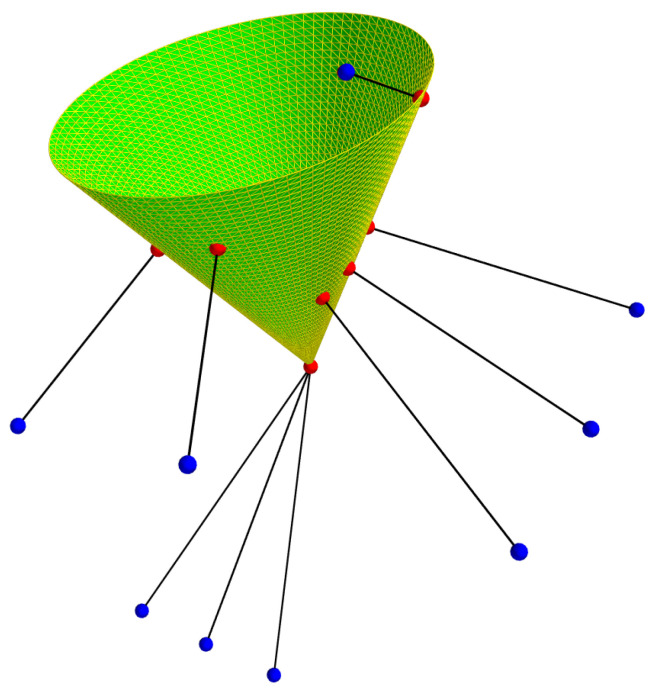
Example showing the closest points on the upper part of the cone (in red) to points outside the surface of the cone (in blue).

**Figure 7 sensors-24-04024-f007:**
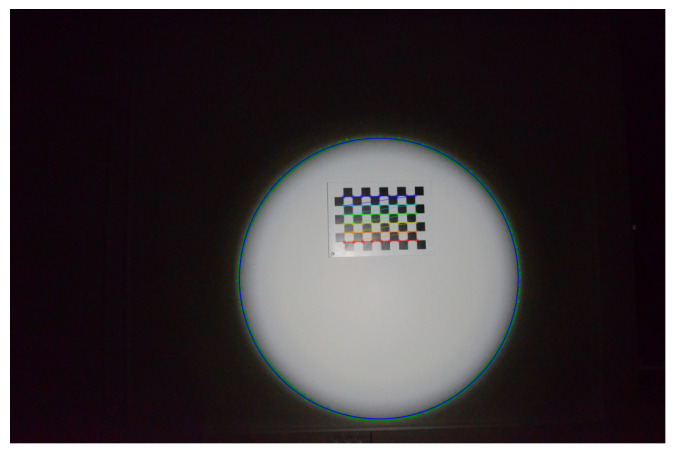
Example of detecting the contour of a halo of light on a wall using a chessboard to estimate the plane pose. The contour is in green and the adjusted ellipse in blue.

**Figure 8 sensors-24-04024-f008:**
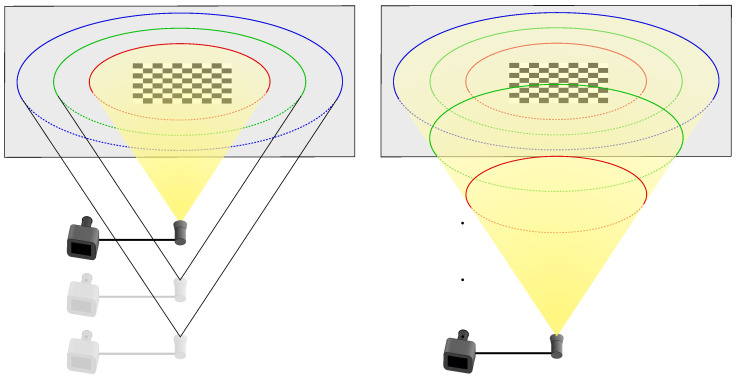
Method for obtaining several elliptical sections of the cone.

**Figure 9 sensors-24-04024-f009:**
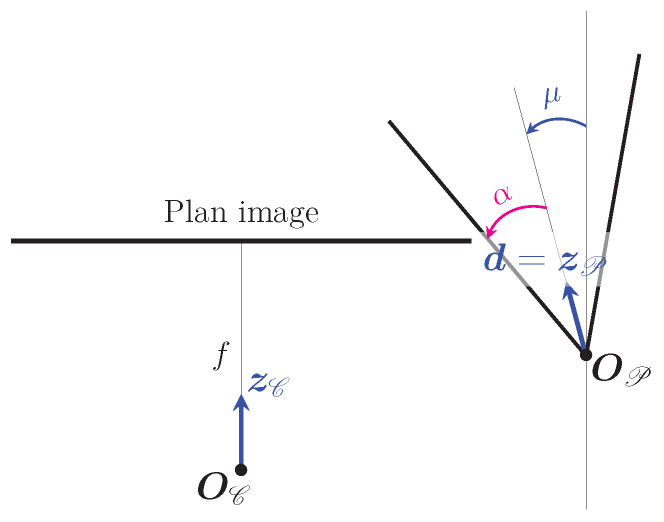
Representation of the camera/cone pair in an orthogonal projection 2D view where the projection axis is perpendicular to $z_{P}$ and $z_{C}$. The relative orientation of the cone with the camera can be defined here by a single angle called μ. This is only true if one considers that the camera has an infinite field of view and is therefore symmetrical about its axis defined by $z_{C}$ (the cone is basically symmetrical about its axis defined by $z_{P}=d$).

**Figure 10 sensors-24-04024-f010:**
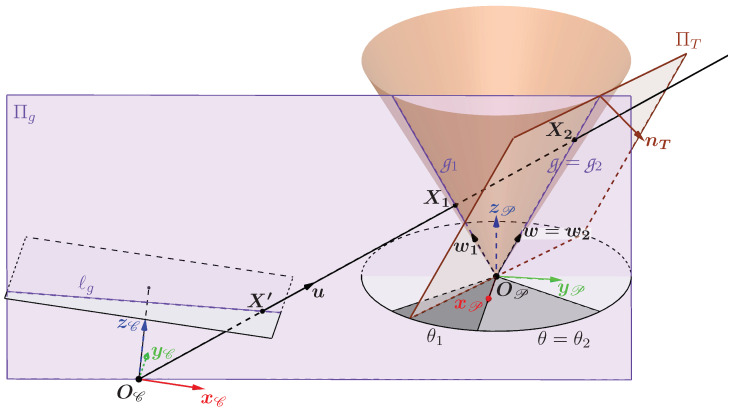
Representation of the plane $\Pi_{g}$ when it is not tangent to the cone passing through the optical center and the two generatrices $g_{1}$ and $g_{2}$.

**Figure 11 sensors-24-04024-f011:**
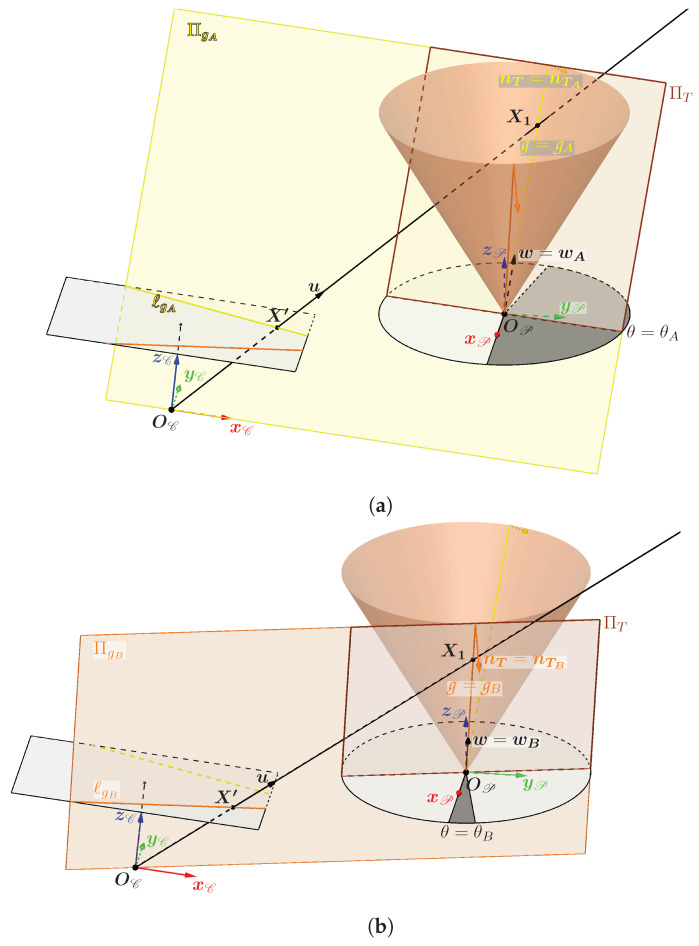
Representation of $\Pi_{g_{A}}$ (**a**) and $\Pi_{g_{B}}$ (**b**), the only two planes tangent to the cone passing through the optical center.

**Figure 12 sensors-24-04024-f012:**
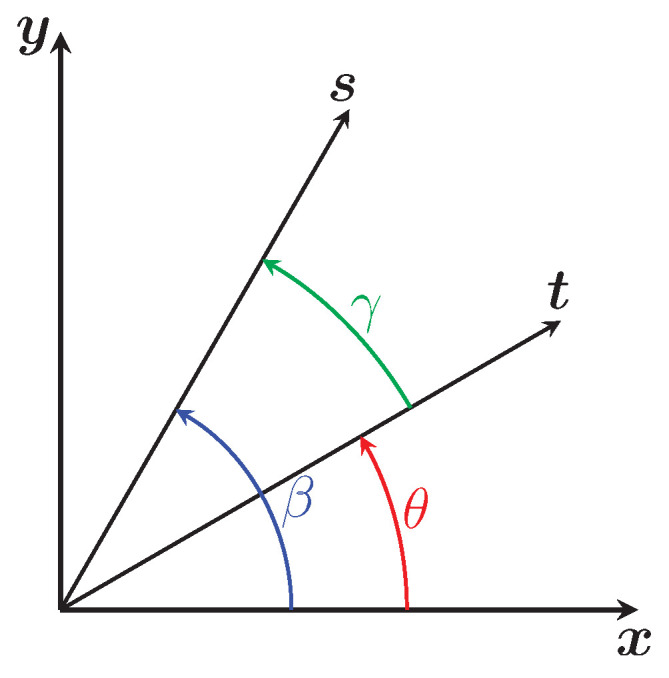
Visualization of the vectors t and s and their angles.

**Figure 13 sensors-24-04024-f013:**
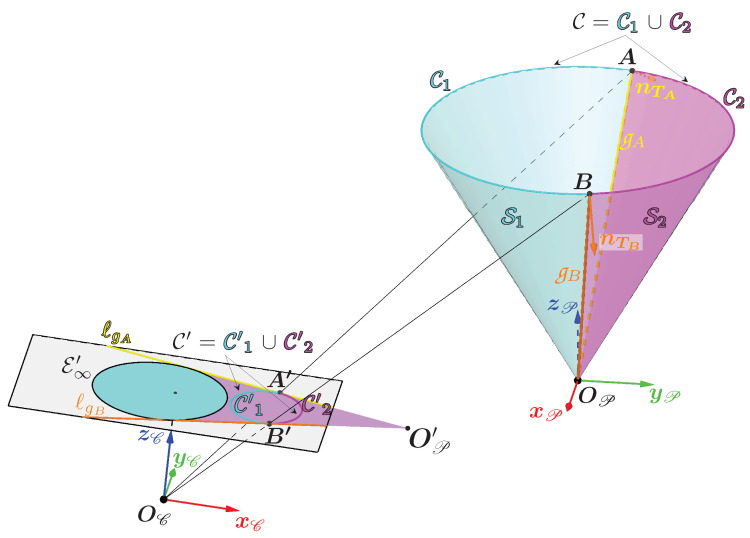
Representation in the image plane and in the cone of the areas containing the first intersections (in cyan) and the second intersections (in magenta).

**Figure 14 sensors-24-04024-f014:**
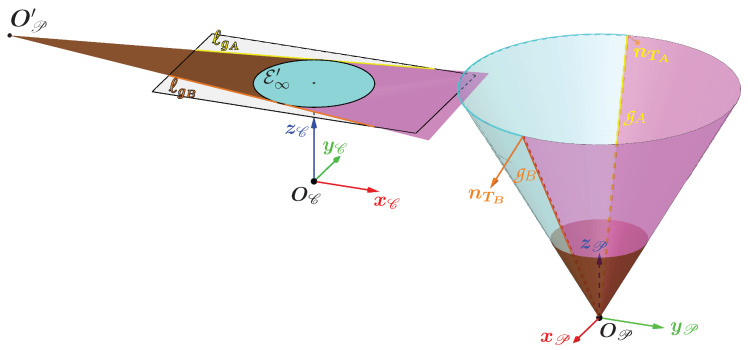
Representation in the image plane and in the cone of the areas containing the first intersections (in cyan) and the second intersections (in magenta) in the case where part of the cone (in brown) is behind the camera.

**Figure 15 sensors-24-04024-f015:**
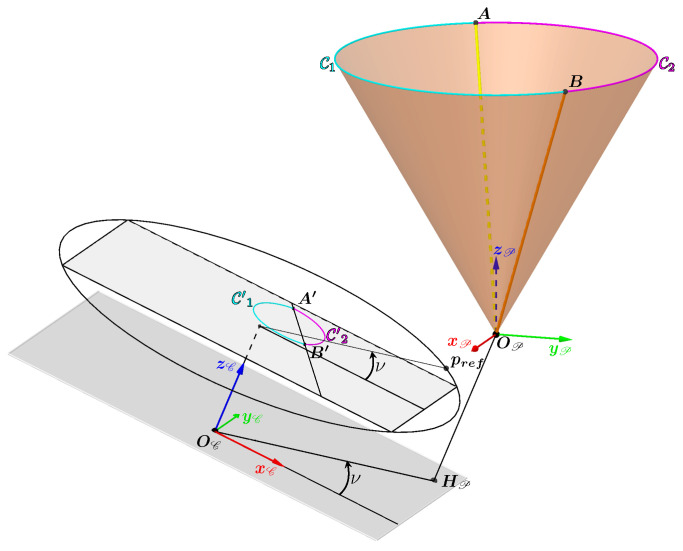
Illustration of how to obtain $p_{\mathrm{ref}}$, the reference point in the image plane which indicates the relative position of the cone, and which is used to obtain the curves ${C^{\prime }}_{1}$ and ${C^{\prime }}_{2}$.

**Figure 16 sensors-24-04024-f016:**
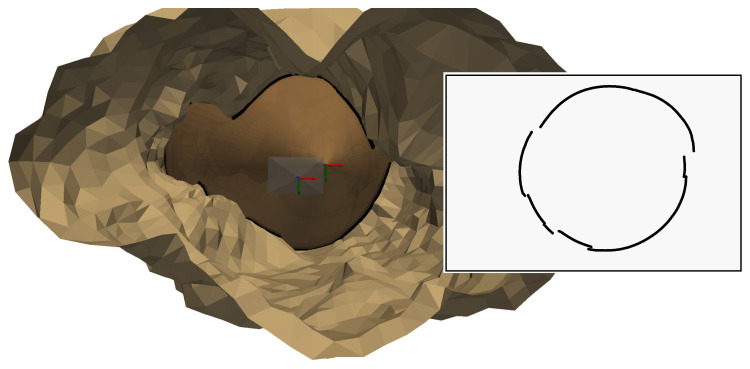
The camera (represented by a pyramid) and the projector (represented by a cone) arranged inside the model of our gallery to simulate a contour of light in an image. This contour is obtained by projecting the intersections between the generatrices of the cone and the model into the image.

**Figure 17 sensors-24-04024-f017:**
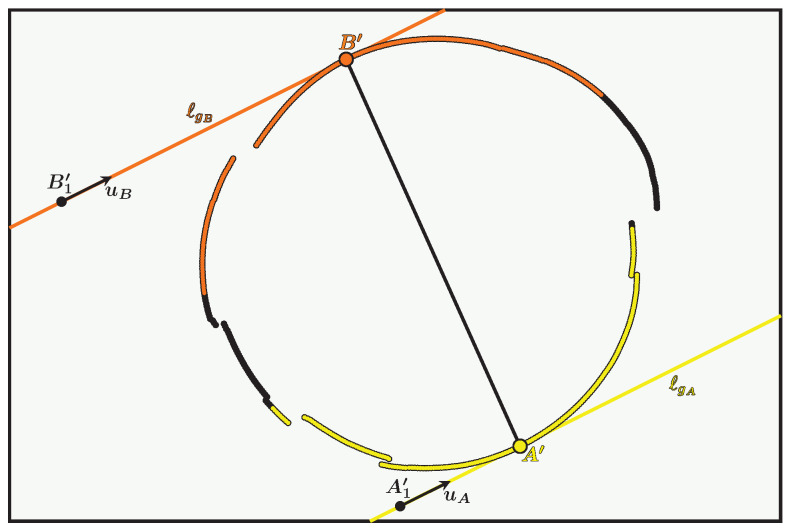
Determining the points $A^{\prime }$ and $B^{\prime }$ for our previous example contour.

**Figure 18 sensors-24-04024-f018:**
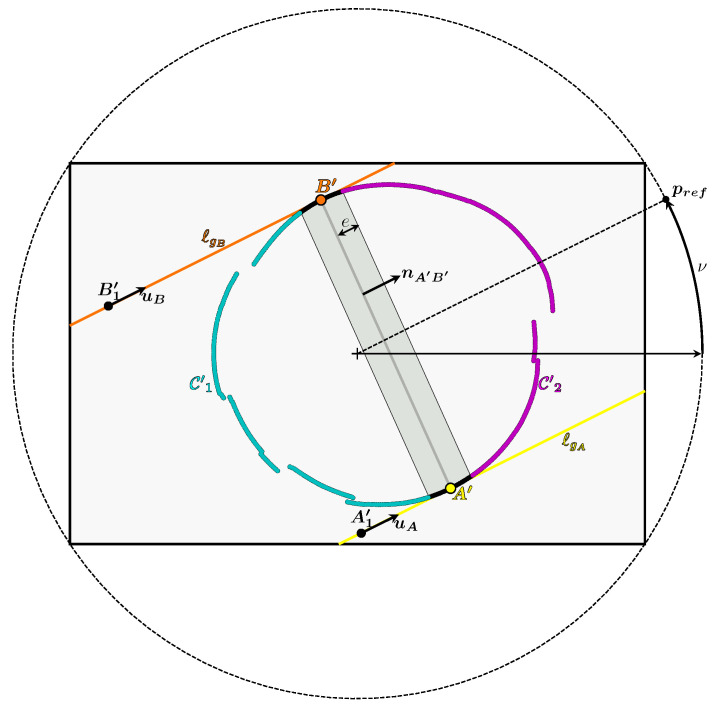
Determining the curves ${C^{\prime }}_{1}$ and ${C^{\prime }}_{2}$ for the previous example contour.

**Figure 19 sensors-24-04024-f019:**
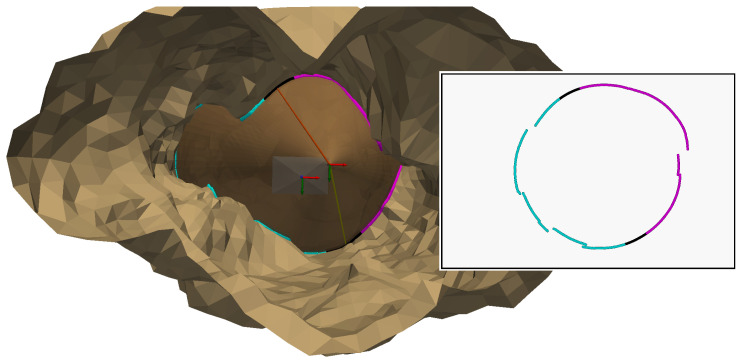
The reconstructed 3D points added to the 3D scene of Figure 19 with the first intersections in cyan and the second intersections in magenta.

**Figure 20 sensors-24-04024-f020:**
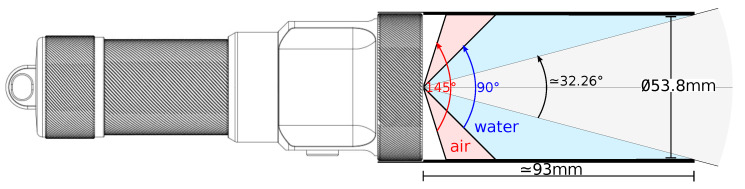
The diving lamp and its angle of aperture.

**Figure 21 sensors-24-04024-f021:**
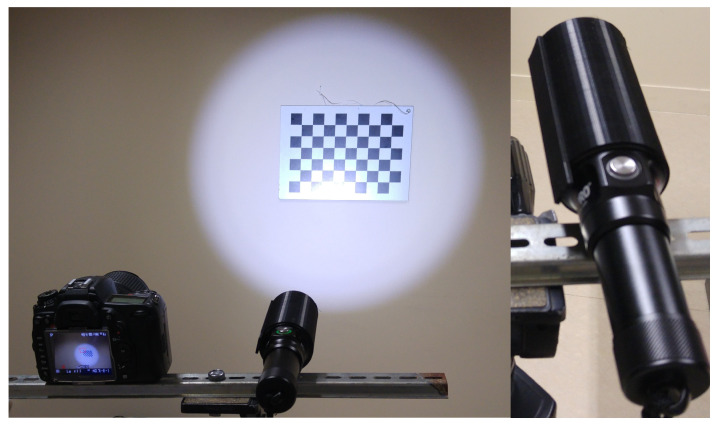
The system used, consisting of a camera and a conical-shaped projector, to which a black cylinder has been added.

**Figure 22 sensors-24-04024-f022:**
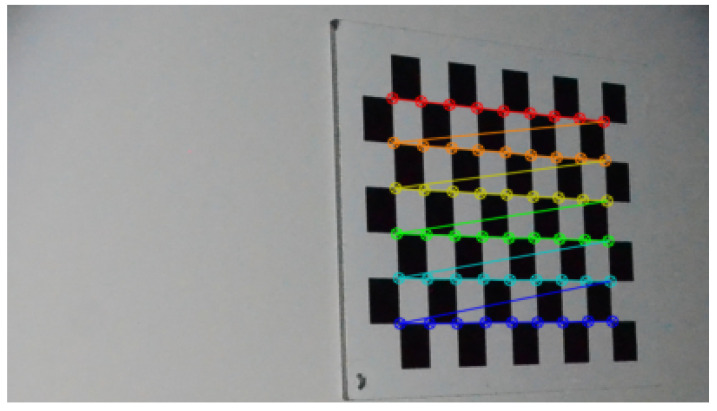
Image of one of the eight chessboard patterns taken at different poses for camera calibration using Zhang’s method.

**Figure 23 sensors-24-04024-f023:**
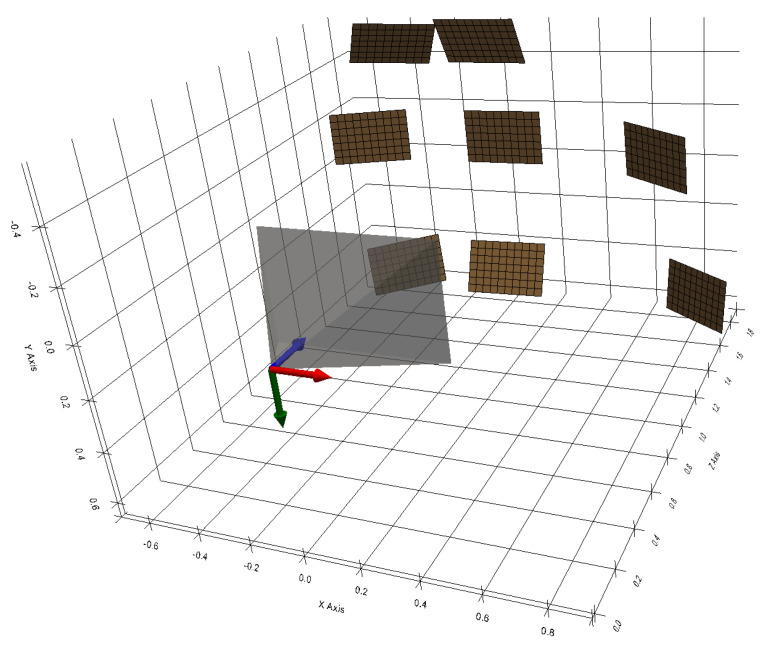
Three-dimensional view of the chessboard in its eight poses in relation to the camera (X, Y, Z).

**Figure 24 sensors-24-04024-f024:**
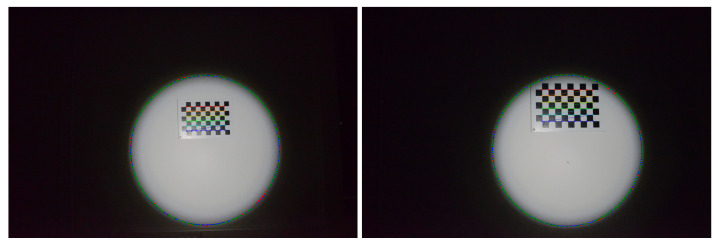
Two images of light projected onto the wall from the five where the axis of the projector is almost orthogonal to the wall (images 1 and 5).

**Figure 25 sensors-24-04024-f025:**
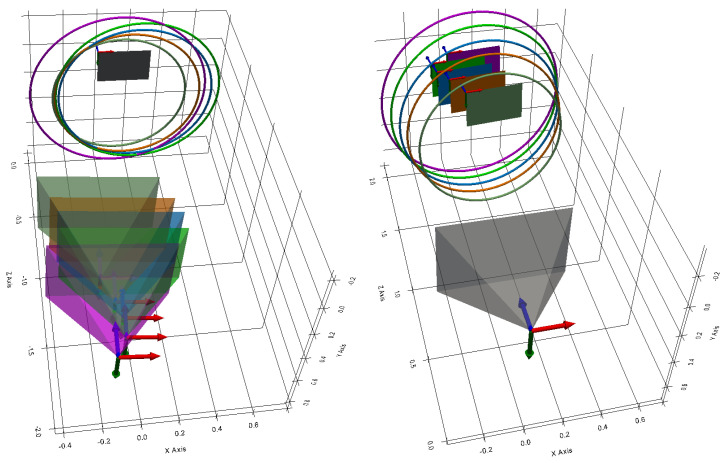
Three-dimensional view of the elliptical sections, the camera and the chessboard pattern. In the image on the left, the elements are expressed in the pattern frame. In the image on the right, the elements are expressed in the camera frame (X, Y, Z).

**Figure 26 sensors-24-04024-f026:**
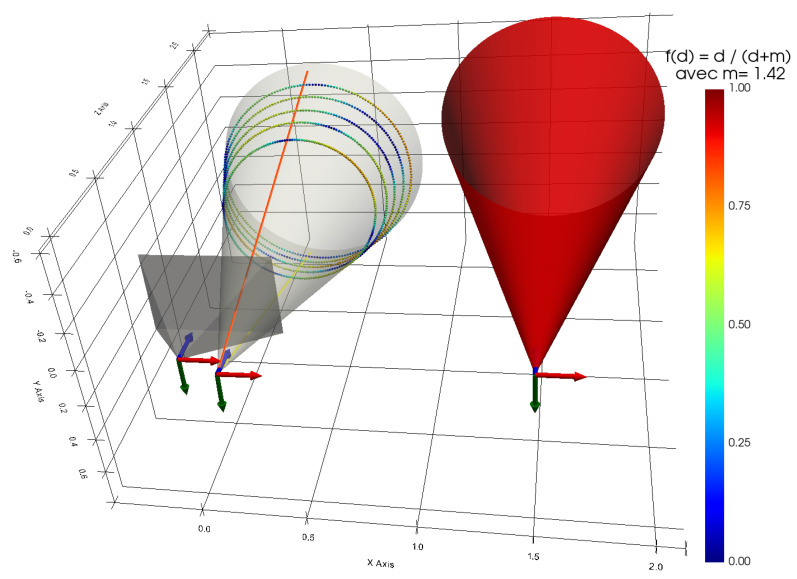
Representation of the estimated cone in relation to elliptical sections where the color of the points depends on the distance orthogonal to the cone (X, Y, Z).

**Figure 27 sensors-24-04024-f027:**
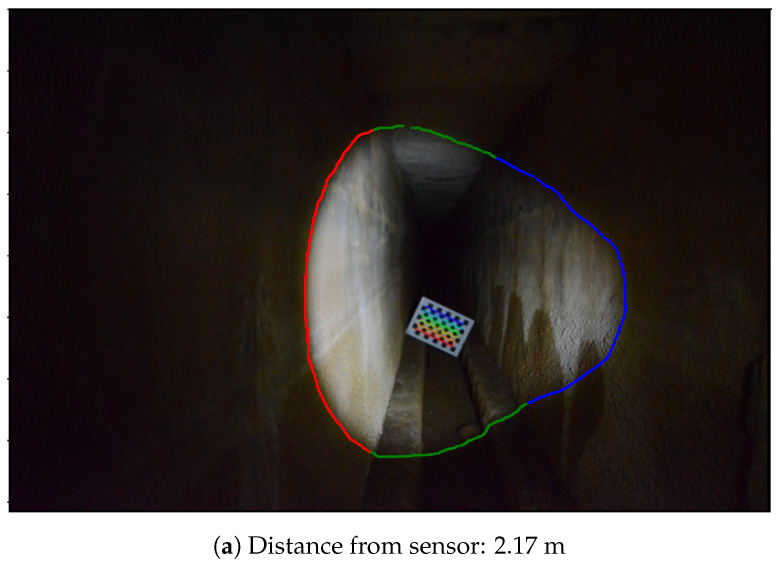
Two images in the aqueduct, at different distances, with light contours.

**Figure 28 sensors-24-04024-f028:**
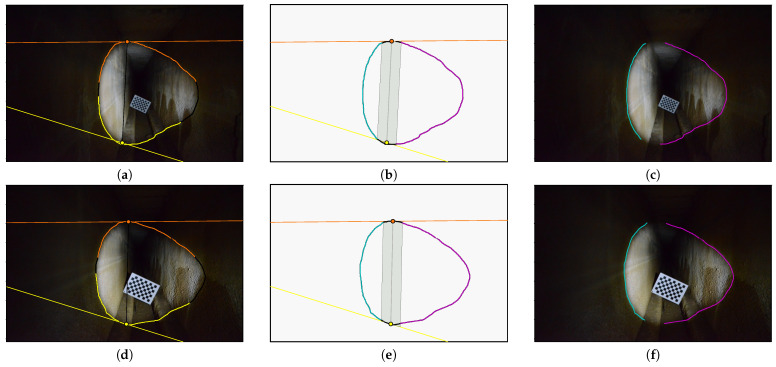
Intersection selection. (**a**) Generatrices projection $\ell_{g_{A}}$, $\ell_{g_{B}}$ (2.17 m). (**b**) Separation of the curves (2.17 m). (**c**) Intersections: left in cyan and right in magenta (2.17 m). (**d**) Generatrix projection $l_{ga}$, $l_{gb}$ (1.50 m). (**e**) Separation of the curves (1.50 m). (**f**) Intersections: left in cyan and right in magenta (1.50 m).

**Figure 29 sensors-24-04024-f029:**
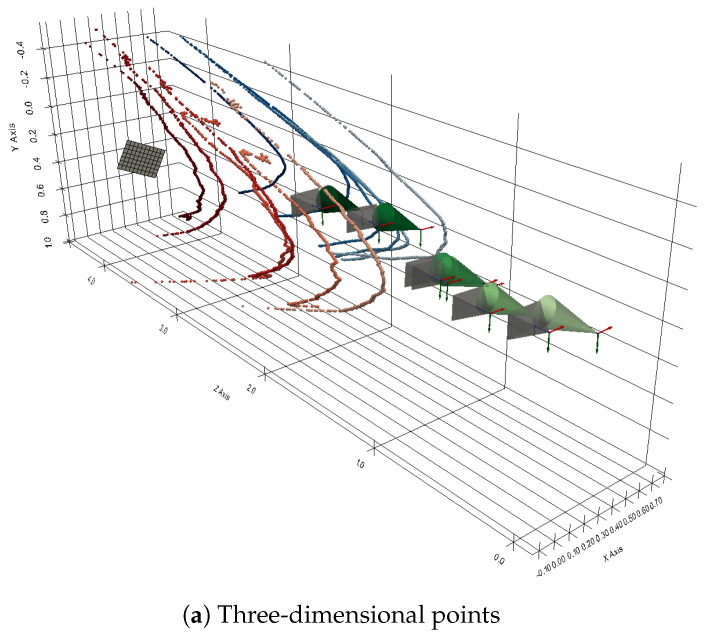
Three-dimensional reconstruction of the contours extracted from the six images (X, Y, Z).

**Figure 30 sensors-24-04024-f030:**
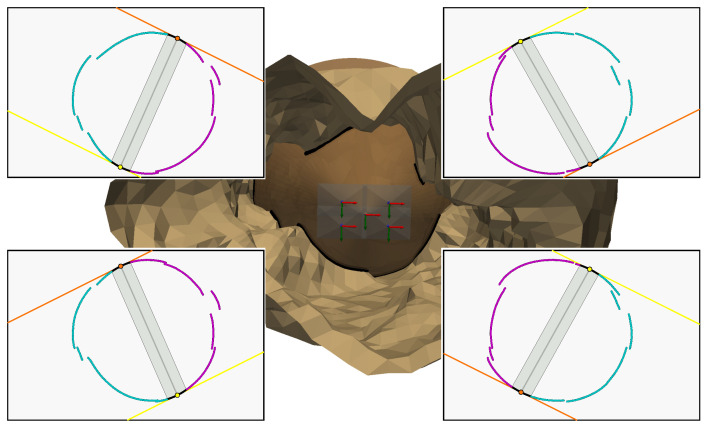
New sensor configuration with four cameras.

**Table 1 sensors-24-04024-t001:** Comparison of estimated and measured cone parameters.

	Estimated Values	Measured Values	Relative Deviation from Measurement
α	14.79°	16.13°	1.34° (−8%)
$∥{O_{P}}_{\left\{C\right\}}∥$	0.210 m	0.212 m	−0.002 m (−0.9%)

**Table 2 sensors-24-04024-t002:** Statistics of the orthogonal distances in mm between the 3D points of each elliptical section and the estimated cone.

Dist. (mm)	Max.	Min.	Med.	Aver.	RMS
Section 0	4.2	0.0	1.4	1.5	2.0
Section 1	3.3	0.0	1.5	1.5	1.8
Section 2	4.1	0.0	1.4	1.6	2.0
Section 3	3.4	0.0	1.6	1.6	1.8
Section 4	3.6	0.0	1.3	1.5	1.8
All	4.2	0.0	1.4	1.5	1.9

**Table 3 sensors-24-04024-t003:** Distance statistics in mm between the 3D points of each elliptical section and the reconstructed 3D points.

Dist. (mm)	Max.	Min.	Med.	Aver.	RMS
Section 0	94.0	0.4	28.5	27.7	32.6
Section 1	149.6	0.0	23.6	30.9	42.8
Section 2	226.1	0.1	18.6	33.6	51.3
Section 3	179.3	0.1	18.6	30.0	44.7
Section 4	114.3	0.1	19.8	23.0	29.1
All	226.1	0.0	20.6	29.0	40.9

**Table 4 sensors-24-04024-t004:** Assessment of coplanarity of 3D points.

(**a**) Orthogonal distances between 3D left points from image *i* and estimated left plane *i*					
**Dist. (mm)**	**Max.**	**Min.**	**Med.**	**Aver.**	**RMS**
image 0	36.4	0.0	2.1	3.6	5.6
image 1	60.1	0.0	4.7	8.0	13.2
image 2	89.9	0.0	9.0	13.3	20.3
image 3	24.6	0.0	5.4	5.7	6.7
image 4	36.1	0.0	5.1	5.8	7.6
image 5	33.6	0.0	3.9	5.6	7.9
(**b**) Orthogonal distances between 3D left points from all images and the global estimated left plane					
**Dist. (mm)**	**Max.**	**Min.**	**Med.**	**Aver.**	**RMS**
All	220.5	0.0	11.0	13.9	19.9
(**c**) Orthogonal distances between 3D right points from image *i* and estimated right plane *i*					
**Dist. (mm)**	**Max.**	**Min.**	**Med.**	**Aver.**	**RMS**
Image 0	27.5	0.0	5.2	6.6	9.0
Image 1	44.2	0.0	18.5	16.6	20.0
Image 2	29.3	0.0	10.1	10.2	12.1
Image 3	21.3	0.0	7.1	7.5	8.8
Image 4	26.7	0.0	3.4	5.7	8.0
Image 5	46.4	0.0	6.5	9.1	12.3
(**d**) Orthogonal distances between 3D right points from all images and the global estimated right plane					
**Dist. (mm)**	**Max.**	**Min.**	**Med.**	**Aver.**	**RMS**
All	113.9	0.0	20.4	24.1	30.5

**Table 5 sensors-24-04024-t005:** Measuring the distance between the two walls.

(**a**) Orthogonal distances between 3D left points from image *i* and estimated right plane *i*					
**Dist. (mm)**	**Max.**	**Min.**	**Med.**	**Aver.**	**RMS**
image 0	641.4	576.6	591.4	593.1	593.2
image 1	622.0	484.1	607.9	596.5	597.1
image 2	757.7	542.6	582.2	592.6	593.3
image 3	596.6	485.5	574.8	567.7	568.2
image 4	598.0	482.0	579.6	569.7	570.3
image 5	619.4	496.7	592.0	581.8	582.5
(**b**) Orthogonal distances between 3D left points from all images and the global estimated right plane					
**Dist. (mm)**	**Max.**	**Min.**	**Med.**	**Aver.**	**RMS**
All	778.4	492.1	581.9	584.9	585.2
(**c**) Orthogonal distances between 3D right points from image *i* and estimated left plane *i*					
**Dist. (mm)**	**Max.**	**Min.**	**Med.**	**Aver.**	**RMS**
Image 0	628.6	581.2	594.0	599.1	599.2
Image 1	602.9	466.0	584.5	573.4	574.4
Image 2	767.6	563.9	573.9	586.2	587.5
Image 3	597.1	508.0	565.2	564.2	564.7
Image 4	625.6	529.5	609.6	597.5	598.1
Image 5	603.2	500.7	578.2	568.9	569.6
(**d**) Orthogonal distances between 3D right points from all images and the global estimated left plane					
**Dist. (mm)**	**Max.**	**Min.**	**Med.**	**Aver.**	**RMS**
All	691.1	476.9	585.2	583.6	584.4

## Data Availability

Data will be available upon request.

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
