# Peer review of "A Novel 3D Reconstruction Sensor Using a Diving Lamp and a Camera for Underwater Cave Exploration"

_sensors, 2024, doi:10.3390/s24124024_

Round 1

Reviewer 1 Report

Comments and Suggestions for Authors

Please find the attached suggestions. I kindly request your focus on revising the materials and methods section and writing a discussion, as this was omitted.

Comments on the Quality of English Language

Please proofread the paper to check for grammatical errors.

Author Response

Dear Reviewer.

First of all, we would like to thank you for your time reviewing our article and for the advice and comments you have given us.

To respond, we have chosen to place comments in front of your remarks.

Hoping that it will suit you, you will find them in the attached file.

A new version of the manuscript can be found HERE.

Regards.

Quentin Massone,

Sébastien Druon,

Jean Triboulet.

Reviewer 2 Report

Comments and Suggestions for Authors

The topic covered in the paper is extremely important, as is digitization for underwater cave exploration. Praise to the authors for this kind of research.

Write in the passive voice. 

In the Introduction, where an overview of the research in the field is given, group the works according to similar topics or methods used. Expand the literature with additional references from journals. 

Chapter 2 is Related works, not Main methods.

Explain what ROVs and AUVs are in line 57.

What is ORB-SLAM? New terms must not be introduced without explanation.

What is "this" triangulation in line 104?

For the title of Figure 2, write that it is a pipeline of the 3D Reconstruction Methods in Underwater Environments instead of a review.

The sequence number of the reference cannot be at the beginning as in line 81. List the names of the first authors et al. and only then the sequence number.

SfM is a photogrammetric approach. Describe to be clear in line 115. For photogrammetric surveying, an accurate shooting plan must be created.

Line 130: For photogrammetric surveying diffuse lighting is necessary.

Images should be displayed immediately below the paragraphs that describe them. In Figure 4, the term frame is used for the global coordinate system, the camera coordinate system, and the projector coordinate system. It is clearer to use the term coordinate system. How are the orientations of these coordinate systems determined in Figure 4? Add orthogonal projection for Figure 4. 

Check the orientations of the coordinate systems in Figures 13-15.

Add orthogonal projection for Figure 23. For example the front view, the blue axis (I assume x ) is positively oriented to the right. 

Figure 24: Number 2 can be written in letters. What are images 1 and 5?

Figure 28 must be shown more clearly and each of its segments explained.

Figure 29: each of its segments must be explained (a and b).

The caption for Figure 30 needs to be more detailed, configuration of what?

No points were made in the Discussion, and the text is problematic because it is generically written. There was no Discussion of the obtained results at all.

Rework the Conclusion so that the research conclusions are presented. Also, in a separate subheading, separate the directions of future investigation.

The Conclusion should not contain a Figure, let it be transferred above, as well as the text describing it, in the results chapter.

A scientific paper should not contain sentences starting with "We also think...." All the above results must be supported by facts and evidence.

Check all the equations once more.

Author Response

(The authors gave the same response as above.)

Reviewer 3 Report

Comments and Suggestions for Authors

The proposed 3D reconstruction method using structured light, offers a low-cost solution for underwater cave exploration with a diving lamp and camera setup.  This work has potential significance in the field of karstic structure research and underwater archaeology and geology, while the methods used are without very novelty. More details issues are as follows:

1. Including quantitative metrics to evaluate reconstruction accuracy, such as mean squared error, mean absolute error, or mean angular error, would strengthen the persuasiveness of your results. I suggest the authors use the standard ball to test the quantitative results.

2. A comparison with existing 3D reconstruction techniques would be useful to demonstrate the advantages of your method in terms of accuracy, cost, and ease of operation.

3. The proposed method was tested in a dewatered environment, but actual underwater conditions can be more complex. Further discussion on the challenges the method may face in real underwater environments is needed. For example,  in underwater environments, the relative positioning of devices may change, affecting the accuracy of 3D reconstruction. Adding the real underwater experiments would be a better choice if possible.

4. In Fig. 2, lack of photometric stereo methods for underwater 3D reconstruction, with references [1,2,3]. 

[1] Underwater optical 3-D reconstruction of photometric stereo considering light refraction and attenuation 

[2] Deep Learning Methods for Calibrated Photometric Stereo and Beyond

[3] Gps-net: Graph-based photometric stereo network

Author Response

(The authors gave the same response as above.)

Round 2

Reviewer 1 Report

Comments and Suggestions for Authors

Well done on the improvements to the paper.

Comments on the Quality of English Language

I have picked up a few grammatical errors. Please read through the manuscript again.

Reviewer 3 Report

Comments and Suggestions for Authors

The authors addressed my issues.